# Extracellular Vesicles in Breast Cancer: From Biology and Function to Clinical Diagnosis and Therapeutic Management

**DOI:** 10.3390/ijms24087208

**Published:** 2023-04-13

**Authors:** Sylvain Loric, Jérôme Alexandre Denis, Cédric Desbene, Michèle Sabbah, Marc Conti

**Affiliations:** 1INSERM U538, CRSA, Saint-Antoine University Hospital, 75012 Paris, France; jerome.denis@aphp.fr (J.A.D.); cedric.desbene@aphp.fr (C.D.); michele.sabbah@inserm.fr (M.S.); marc.conti@integracell.net (M.C.); 2INTEGRACELL SAS, 91160 Longjumeau, France

**Keywords:** extracellular vesicles, exosomes, breast cancer, diagnostic, prognosis, therapy, targeting, vector

## Abstract

Breast cancer (BC) is the first worldwide most frequent cancer in both sexes and the most commonly diagnosed in females. Although BC mortality has been thoroughly declining over the past decades, there are still considerable differences between women diagnosed with early BC and when metastatic BC is diagnosed. BC treatment choice is widely dependent on precise histological and molecular characterization. However, recurrence or distant metastasis still occurs even with the most recent efficient therapies. Thus, a better understanding of the different factors underlying tumor escape is mainly mandatory. Among the leading candidates is the continuous interplay between tumor cells and their microenvironment, where extracellular vesicles play a significant role. Among extracellular vesicles, smaller ones, also called exosomes, can carry biomolecules, such as lipids, proteins, and nucleic acids, and generate signal transmission through an intercellular transfer of their content. This mechanism allows tumor cells to recruit and modify the adjacent and systemic microenvironment to support further invasion and dissemination. By reciprocity, stromal cells can also use exosomes to profoundly modify tumor cell behavior. This review intends to cover the most recent literature on the role of extracellular vesicle production in normal and cancerous breast tissues. Specific attention is paid to the use of extracellular vesicles for early BC diagnosis, follow-up, and prognosis because exosomes are actually under the spotlight of researchers as a high-potential source of liquid biopsies. Extracellular vesicles in BC treatment as new targets for therapy or efficient nanovectors to drive drug delivery are also summarized.

## 1. Introduction

Breast cancer (BC) is the first worldwide most commonly diagnosed cancer for both combined sexes and accounts for 11.7% of total cancer incidence and 6.9% of cancer-related deaths. In 2020, more than 2.3 million new BC cases were reported worldwide, with almost 685,000 related deaths, according to WHO. The death rate was considerably higher in developing versus developed countries (15.0 vs. 12.8 per 100,000) [1]. In Europe and the USA, approximately 523,000 and 276,500 new BC cases are diagnosed annually [2], while 138,000 and 42,000 die of BC-related diseases. Although BC mortality has been thoroughly declining over the last decades, there are still large differences between women diagnosed with early BC (considered curable with ∼96% 5-year survival probability in Europe) or when metastatic BC is diagnosed (mostly not curable with ∼38% 5-year survival rate) [3]. The two significant supports of BC management are locoregional treatment and systemic therapy, and both histological and molecular characteristics of BC broadly impact treatment choices.

Nevertheless, even with the development of new efficient therapies (immunotherapy with PDL1 inhibitors), BC recurrence and metastasis are still the leading causes of death [4], mainly because of the emergence of therapy-resistant cancer cells [5,6]. Thus, answering essential questions involving the factors or mechanisms determining the distant metastasis or the acquisition of therapy resistance is strongly mandatory to develop novel effective therapeutic strategies against BC [7]. Recent studies have shown the importance of tumor evolution in the continuous interplay between tumor cells and their microenvironment (cancer cells’ interaction with themselves, bidirectional interaction of cancer cells with stromal cells, etc.) [8,9]. Such communication strategies require specific mechanisms, including direct cell-to-cell contacts but also autocrine, juxtacrine, paracrine, and even endocrine secretion of specific factors (growth factors, matrixins, cytokines, and chemokines) [10]. Among such secreted means are figure exosomes [11], a generic consensus term used to describe any small lipid bilayer-delimited particles that are unable to replicate and are extracellularly released by every cell (including prokaryotic ones) [12,13]. Exosome surface receptors allow their targeting and capture by a broad range of recipient cells that will incorporate either proteic, lipidic, or genetic messages resulting in modifications of their physiological behavior. These exosomes have recently been proven to be efficient means of communication in human diseases [14], especially in cancer [15,16]. As the field of exosomes is highly active [17,18,19], we aimed to review the respective roles of cancer cell-derived exosomes as well as stromal-derived exosomes in BC to better understand the cellular and molecular mechanisms underlying their generation and development. We also emphasize exosomes as powerful tools to efficiently diagnose, better stage and improve BC prognosis, and better design, in a personalized approach, treatment strategies.

## 2. Extracellular Vesicle Nature, Structure, and Properties

Various amounts of 40–1000 nm membrane-derived extracellular vesicles, continuously released from the plasma membrane (plasma membrane) into the local environment by either eukaryotic or prokaryotic cells, can be detected in almost all biological fluids. The 2018 MISEV consensus rule recommends differentiating label bilayered membrane vesicles such as SEVs (smaller than 200 nm, small extracellular vesicles or exosomes) and MLEVs (larger than 200 nm, medium large EVs or ectosomes) shed by live cells [20,21] from apoptotic bodies or necrotic blebs of a plasma membrane that are the consequences of dying cells disassembly [22].

### 2.1. Extracellular Vesicle Biogenesis

While MLEVs are heterogeneous membranous vesicles generally deriving from outward plasma membrane budding (ectosome release) [23], SEVs originate from the endosomal compartment [24].

#### 2.1.1. SEVs Biogenesis

SEVs production follows complicated sorting routes and requires several complex protein systems [25,26,27]. SEVs biogenesis starts with the inward budding of small portions of the plasma membrane containing an outer membrane exposed material. These small intracellular vesicles form the early endosome, which subsequently matures and transforms into a late endosome. Inward budding of the limiting membrane of the late endosome then occurs, resulting in the progressive assemblage of intraluminal bilayered vesicles (ILVs) within so-called large multivesicular endosomes or multivesicular bodies (MVBs) (Figure 1). At this step, it seems that the endocytosed cargo is first delivered to the trans-Golgi network (cargo-in) and then back transferred to MVB (cargo-out) [28]. This route seems to follow the MVB-driven Golgi protein quality control pathway that will further degrade miss-folded proteins in endolysosomes [29]. During this process, cytosolic proteins, as well as nucleic acids, can be trapped into ILVs through the action of the ESCRT (endosomal sorting complex required for transport) machinery [30,31] or by following a ceramide ESCRT-independent pathway, suggesting a critical role for lipid raft microdomains in MVBs formation [32]. ESCRT is a family of proteins that associate in successive complexes (ESCRT-0, -I, -II, and -III) at MVBs membrane to sort ubiquitinated cargos into late endosomes [33].

The fate of MVBs varies according to the proteins that are expressed on their surface that will be specifically recognized on the target membrane [34]. Acidification along the endocytic pathway also seems to be required for the degradation and recycling of internalized components [35]. Intracellular calcium [36] and local hypoxia [37] also seem to be major determinants for MVBs degradation versus secretion. Most MVBs are directed for cargo degradation into lysosomes by fusing with them [38,39]). It was demonstrated that the autophagosome may fuse with MVB in a pre-lysosomal step, resulting in a hybrid organelle called amphisome [40], suggesting that both the autophagy degradation process and the exosomal release are closely linked [41]. As MVBs also contain intraluminal proteins and lipids, which are not intended for lysosome degradation, ILVs can release their content into the cytoplasm by direct back-fusion with the endosome-limiting membrane [42]. However, a subset of them fuse to the plasma membrane and release their content externally in the form of SEVs, a secretion process called exosome biogenesis [43,44].

MVBs that are fated for exocytosis are transported to the plasma membrane along cytoskeletal structures such as actin or microtubules and their associated molecular motors kinesins and dyneins [45,46]. MVBs docking to the plasma membrane is strongly regulated by the Rab family of small GTPases proteins [47], especially Rab27a which mediates the docking, tethering, and fusion of MVBs with the plasma membrane [48,49]. Once docked, secretory MVBs couple to the SNARE (soluble N-ethylmaleimide-sensitive component attachment protein receptor) membrane fusion machinery [50]. SNARE complex formation and membrane fusion are tightly controlled by multiple regulatory mechanisms [51], which include the phosphorylation profile of SNARE proteins, which influence either SNARE complex localization or interaction with SNARE partners [52]. SNARE assembly and disassembly mediate the fusion of MVBs with the cell membrane, thus releasing outside the cell the MVBs particles [53] that become exosomes.

#### 2.1.2. MLEVs Biogenesis

Interestingly, in contrast with the importance of MVBs in the SEV formation pathway, MLEV release is totally MVB-independent and does not require exocytosis [54,55]. MLEV are assembled by the regulated outward budding of plasma membrane domains [56], a mechanism depending either on caveolae [57] or clathrin-coated vesicles [58], explaining why ectosomes are surrounded by phospholipid membranes containing lipid rafts and caveolae [59].

However, despite the distinct mechanism for biogenesis and membrane origin, both endosome-origin SEVs and MLEVs can work similarly, and the crucial difference between them has not yet been elucidated [23].

### 2.2. Extracellular Vesicle Composition

MLEVs, which bud directly from the plasma membrane of healthy cells, contain cytoskeleton and endoplasmic reticulum elements [60,61]. It is considered that MLEVs’ composition mainly reflects the surface proteins of parental cells [23]. As phosphatidylserine repositioning within the cell plasma membrane is a critical factor in MLEVs evagination [62], MLEVs display high levels of phosphatidylserine. In contrast, SEVs have lower ones exposed to the outer membrane leaflet [63].

Because of their endosomal origin, and since they derived from the ILVs in MVBs, SEVs biogenesis heavily depends on the mechanisms that regulate MVBs maturation and trafficking. Along the different sorting mechanisms needed for SEVs production, specific molecules (proteins, lipids, amino acids, metabolites, nucleic acids such as nuclear or mitochondrial DNA or several RNA species, etc.) [64,65] are incorporated into SEVs generating cargo diversity [66,67,68].

SEVs mostly contain proteins originating from the cytosol, the endosomal compartment, and the plasma membrane [12]. Cytosolic protein engulfment involves proteins close to the MVB outer membrane during its inward budding. Proteins such as Heat shock proteins (Hsp90, Hsc70, Hsp60, Hsp20, Hsp27, etc.), growth factors and cytokines (TNF-α, TGF-β, TRAIL, etc.), and enzymes (belonging to central metabolisms such as glycolysis, citric acid cycle, etc.) can be found in exosome lumen [69]. As the budding and release of SEVs require inner plasma membrane actin polymerization, and then the actomyosin cytoskeleton contraction, cytoskeleton proteins such as actin, actinin, dynamin, myosins, and tubulin are also generally found in SEVs [45]. It holds the same for essential regulators of extracellular vesicles trafficking: ESCRT complex proteins and important ESCRT partners molecules implicated in ESCRT assembly or nucleation such as ALIX [70], members of the Rab family [71], and SNARE membrane fusion machinery, required explicitly for MVBs docking and fusion with plasma membrane [72], are also found in SEVs (Figure 2). Tetraspanins (mainly CD9, CD63, CD37, CD81, CD82, and CD53), which are highly conserved integral membrane proteins displaying a high affinity for cholesterol and sphingolipids such as ceramides, are involved in ESCRT-independent exosome release [73] and greatly influence exosome biogenesis and composition [74,75]. They play essential roles in plasma membrane protein scaffolding and anchoring in cellular membranes by creating specific plasma membrane tetraspanin-enriched microdomains [76]), thus facilitating their sorting into SEVs [77,78]. Thus, antigen-presenting molecules (Major Histocompatibility complex MHC class 1 and MHC class 2), glycoproteins (O-linked and N-linked glycans), adhesion molecules (integrins, selectins, etc.), and signaling receptors (TNF receptor, Transferrin receptor, etc.) are frequently found on SEVs membrane [79].

SEVs can also comprise nucleic acid molecules [80]. Various RNA species (mRNAs, rRNA, tRNA, snRNA, snoRNA, piRNA, Y-RNA, scRNA, SRP-RNA, 7SK-RNA, miRNAs (miRs), lncRNAs, circRNAs, etc.) can be evidenced in SEVs [81,82]. Numerous reports have shown the ability of SEVs RNAs to profoundly impact the functional properties of cells that incorporate them [81]. Nuclear and mitochondrial DNA molecules can also be conveyed by SEVs [83,84]. In plasma, cell-free DNA (CFDNA) circulates in both free form and enclosed in SEVs [83]. While large intact DNA is generally associated with MLEVs [85] and is mainly attached to the outer surface of extracellular vesicles [86], shorter double-strand 150 to 6000 bp fragments resulting from DNA fragmentation by DNAses are usually found in SEVs [87]. As CFDNA sometimes harbors mutations, it may reflect the mutational status of parental DNA [88] and serve as a relevant tumor biological marker in liquid biopsies [89,90]. Aside from this complex protein and nucleic acid repertoire in SEVs, metabolomic studies reveal that SEVs contain different classes of low-molecular-weight compounds.

Organic acids, nucleotides, sugars, their derivatives, carnitines, vitamins, related metabolites, and amines are frequently evidenced in SEVs [91]. Lipids (phosphatidylserine, cholesterol, sphingomyelins, and ceramide) participating in intercellular signaling and also ensuring structural stability are present [92]. These metabolites may originate from specific sorting but are more probably synthesized in situ in SEVs as complete, but more often, partial metabolic routes can be evidenced [93].

Extracellular vesicles, released in body fluids, vastly differ in size, content, morphology, and biological mechanisms [94,95]. A single cell line can continuously generate morphologically diverse vesicles [96]. However, little is known about the essential mechanisms that may account for the combinatorial repertoires of SEV cargo and the heterogeneity in cargo compositions across different extracellular vesicle populations and subtypes [97]. Many reports using either proteomics, transcriptomics, or metabolomics have shown how the vesicular protein cargo is distinct from its original sample [98,99,100]. Increasing evidence has pointed to the selectivity in cargo loading during SEV biogenesis rather than a generic regulation of cargo sorting into SEVs [101]. Under hypoxic conditions, tumor cells show changes in morphology, distribution, and accumulation cargo of MVBs. These modifications are associated with significant differences in the number, morphology, and cargo of SEVs [34]. It was also reported that polymorphonuclear neutrophil cells could produce a broad spectrum of SEVs, depending on the environmental conditions prevailing during SEV genesis [102]. Consequently, SEV composition does not simply represent parental cell protein composition, but more notably, SEVs cargo multifaceted variety reflects a specific signature of these source cells at a definite time [103,104].

SEV biogenesis threshold will vary significantly between cell types according to their physiological/pathological status. The high rate of SEVs secretion found in transformed cells suggests that the balance between MVB degradation and secretion is disrupted in cancer toward SEVs cargo release [105]. Such modification is not specific to cancer cells but may also occur in non-transformed ones. In antigen-presenting cells, large amounts of SEVs are found to be released upon stimulation [106].

### 2.3. Extracellular Vesicle Fate

Once released, SEVs circulate locoregionally or distantly to deliver their cargo content to recipient cells. The encapsulated cargo of SEVs is protected from degradation [79]. Circulating labeled SEVs’ half-life has been evaluated in mice to be about 2 minutes, but detecting SEVs in the bloodstream remains possible several hours after injection [107]. SEVs then use their lipid membranes to enter recipient cells to release cargo. Although a non-specific uptake is shared by every cell type [108], specific targeting to recipient cells is generally required to deliver exosome cargo and exert its function [109]. When reaching the target cell, SEVs can either trigger signaling by directly interacting with extracellular receptors or be uptaken by direct fusion with the plasma membrane or get internalized. Most reports indicate that endocytosis generally internalizes SEVs into the endosomal compartment [110], while the exact mechanisms underlying SEV endocytosis processes remain unclear [111]. Various other mechanisms have also been proposed, including clathrin-mediated endocytosis, caveolin-dependent endocytosis, lipid raft-dependent endocytosis, micropinocytosis, and phagocytosis [112]. After internalization, SEVs can interact within the recipient cell, inducing intracellular signaling and changes to molecular processes that may affect various functions such as apoptosis, autophagy, growth, cell cycle, migration, invasion, and differentiation [113,114].

## 3. Extracellular Vesicle’s Role in Normal Breast Tissue

### 3.1. Extracellular Vesicles Production in Normal Mammary Tissue

The mammary gland is one of the very few organs in which substantial development occurs only after birth, undergoing cycles of growth, differentiation, milk secretion, apoptosis, regression, and remodeling during the lifetime of the organism [115]. It develops predominantly during the postnatal period from several invading cells derived from the ectoderm [116]. These cells undergo a morphogenetic program that leads to the development of a series of branching ducts that terminate in sac-like lobules embedded in stromal tissue. Both secretory acini and ducts are lined by an epithelium [117] that later expands to generate a complicated network to deliver milk to newborn progeny. This continuous epithelium consists of an outer basal layer of contractile myoepithelial cells and an inner layer of luminal cells surrounding the lumen. The epithelium includes a subset of stem cells closely interacting with the environment to drive their fate and the ultimate mammary gland phenotype [118,119]. The surrounding microenvironment comprises many different cell types that play specific roles in this complex functional network. The microenvironment accounts for nearly 80% of the breast volume and comprises an extracellular matrix and stromal cells including inflammatory cells, adipocytes, endothelial cells, and fibroblasts [120]. In the non-pregnant state, the mammary gland looks like a network of epithelial ducts that empty into the main lactiferous ducts. Epithelial cells’ secretory granules exocytosis releases several antimicrobial peptides inhibiting bacterial growth in the duct system [121]. With pregnancy, and because it must prepare for lactation, the epithelium markedly proliferates and differentiates. It expands to fill the gland, replacing the fat pad with milk-producing lobuloalveolar structures. When milk secretion stops, the mammary gland undergoes apoptosis of the lobuloalveolar cells generated during pregnancy and returns to its original ductal state. Milk SEVs containing MFG-E8 (milk fat globule-EGF (epidermal growth factor)-8) play an essential role in the recognition and engulfment of apoptotic epithelial cells by the neighboring phagocytic cells in the involuting gland [122]. As the epithelial cells are lost, the gland repopulates with adipocytes.

Stromal ECM, which mainly contains type I collagen, fibronectin, laminins, and glycoproteins, is a structural scaffold that maintains breast tissue integrity [123]. Fibroblasts regulate ECM deposition and differentiation of the neighboring epithelium [117,124]. Cell-matrix and cell-cell interactions play critical roles in developing the normal mammary gland, where SEVs can participate [122,125,126]. In the normal gland, SEVs regulate epithelial cell polarity as mammary epithelial cells are highly polarized [127,128]. Several studies have shown that epithelial SEVs that shed apically or basolaterally differ in cargo composition or concentration [129,130]. Distinct loading mechanisms for apical versus basolateral cargo have been suggested [131]. Therefore, polarized secretion of SEVs allows targeted delivery of specific SEV populations to stromal recipient cells due to the organized tissue architecture [128].

### 3.2. Exosome Role in the Maintenance of the Mammary Stem Cell

During female mammals’ sexually active life, the mammary gland continuously undergoes tissue remodeling [132]. During each cyclic pattern of ovarian activity, breast cells proliferate and form alveolar buds at the tertiary side branches, then regress in an ordered fashion [133]. Under pregnancy stimuli, lobuloalveolar differentiation takes place with breast epithelial expansion, which generates complex milk-secreting alveolar units, whose cells undergo terminal differentiation into specialized secretory cells in late pregnancy. Such a dynamic structure with high regenerative capabilities has suggested the existence within the breast of the renewable stem cell population [134]. Mammary stem cells (MaSCs) have been isolated and shown to be able to individually regenerate an entire mammary gland within six weeks in vivo while simultaneously executing up to ten symmetrical self-renewal divisions [135,136]. Localization of dormant MaSCs to the fat pad’s proximal region may indicate a specific microenvironment that resembles the MaSC niche [137]. Stromal fibroblasts appear to be a significant determinant of development in the mammary gland. Several fibroblast-derived factors have been implicated in transmitting signals to the epithelium, including morphogen ligands such as hedgehog or Wnt molecules [138,139]. Mutations within these key-signaling pathways can deregulate MaSCs from controlling regulatory signals, allowing them to develop precursor lesions [140]. Stromal SEVs can also participate in that regulation. Generally, mammary luminal cells do not have stem cell properties and cannot generate mammary glands when implanted into fat pads. SEVs derived from stem-like mammary basal cells can transfer mammary gland-forming abilities to luminal mammary epithelial cells [126]. Such SEVs release is regulated mainly by the presence of SEVs in the extracellular environment [105]. It has been shown to impose quiescence on residual hematopoietic stem cells in the leukemic niche [141].

### 3.3. Milk Is an Essential Source of Extracellular Vesicles

Breast milk, the most important nutritional source for infants, has many beneficial effects. It is rich in various nutrients and ingredients, including proteins, fats, carbohydrates, minerals, and vitamins, which can provide the energy necessary for growth and development in infancy [142]. It also contains many extracellular vesicles whose bilayered structure allows them to remain stable in the baby’s stomach until further absorption by intestinal cells [143]. Once absorbed, maternal EVs can enter the bloodstream and then infant tissues [144], where they will play different vital roles (for review [145]). They will have positive effects on the developing immune system [146,147] and play a role in metabolic regulation [148] and neural development of the newborn [149].

It is well known that pregnancy increases BC risk for all women for at least five years after parturition [150]. In such a context, breast milk extracellular vesicles may be necessary for BC as they may influence BC risk [151]. By promoting epithelial-mesenchymal transition (EMT), milk extracellular vesicles can increase the aggressiveness of both benign and malignant breast epithelial cells when the breast is remodeling, and the surrounding microenvironment is likely to be tumor-promotional [152]. Milk from healthy lactating women contains high levels of TGFβ2 in SEVs that have been evidenced to promote EMT, modifying both MCF7 breast cancer and MC10A breast benign cell lines morphology by disrupting cell-cell junctions and increasing filopodia formation [151].

## 4. Extracellular Vesicles Deregulation in Breast Cancer

### 4.1. Extracellular Vesicle and Cancer Stem Cells

Tumor initiation, therapeutic resistance, recurrence, as well as metastasis have also been associated with the concept of stemness and plasticity in BC [6,153,154]. A relatively rare self-renewal sub-population may drive epithelial cancers, multipotent cells, cancer stem cells (CSCs), or tumor-initiating cells (TICs). Unlike normal adult stem cells that remain constant in number, such cells can increase as tumors grow and give rise to progeny that can be either locally invasive or colonize distant metastatic sites.

As for any other adult stem cells, the properties of mammary stem cells (MaSCs) make them probable candidates for breast cancer initiation [155]. Self-renewal and asymmetric division are stem cells’ cardinal properties that are tightly regulated within the MaSC niche [156] and confer to MaSCs both preserved replicative capacity and resistance to differentiation. Indeed, MaSCs stemness acquisition occurs through the initiation of an epithelial-mesenchymal transition program that activates primary ciliogenesis, which then enables Hh signaling [157]. Mutations in such complex regulatory systems may induce the development of mammary TICS (MaTICs) [158,159] and the MaSCs neoplastic counterpart [160]. MaTICs that have already undergone epithelial–mesenchymal transition possess motility characteristics and can spread in foreign tissues to form a metastatic mass. Thus, MaSCs can harbor mutations over a prolonged life span [140], allowing them to be the true site of breast cancer initiation [161].

MaTICs represent a potential source of tumor heterogeneity. Their high plasticity associated with the random nature of mutations confers variable properties contributing to the considerable cellular heterogeneity observed in human breast cancers [162]. Either MaSCs or MaTICs share the capability to cross-communicate with their environment to maintain homeostasis. It allows the generation of mature breast functional cells throughout life without depleting the pool of stem cells [135,163,164]. The overabundance of microenvironmental stimuli received by the stem cell niche can support the observed phenotype MaTIC variability [165]. Aside from the numerous factors that can modulate the persistence of quiescent/slow-cycling cells in the niche, SEVs transfer figures [166]. Every tiny variation or modulation in SEVs delivery during the continuous crosstalk between CSCs and their surrounding microenvironment is critical and could induce significant deregulation and further tumor progression [167]. For example, PGE2/EP4 signaling controls the homeostasis of MaSCs through SEVs release regulation. MaSCs reprogramming can result from EP4-mediated stem cell property SEVs transfer between mammary basal and luminal epithelial cells [168]. MiR-130a-3p inhibits migration and invasion of MaSCs by regulating Rab5B [169]. Chemotherapy-induced BC cells secrete multiple SEVs miRNAs, including miR-9-5p, -195-5p, and -203a-3p, simultaneously targeting the transcription factor One Cut Homeobox 2 (ONECUT2), which induce CSC traits to BC cells, which has been associated with tumor refractoriness and progression [170]. In addition, BC cells prime mesenchymal stromal cells to release SEVs containing miRNAs such as miR-222/223, promoting dormancy in a subset of BC cells and conferring drug resistance [171]. Understanding the importance of SEVs transfer in that context is a crucial feature for future BC therapy [172].

### 4.2. Bidirectional Contributions of Extracellular Vesicles from Breast Tumor and Microenvironmental Cells to Breast Cancer Changes

The tumor microenvironment (TME) is a complex and dynamic network including cancer and stromal cells. Stress conditions such as hypoxia, starvation, and acidosis increase tumor cells’ SEVs release, leading to TME changes and expansion. Such specific behavior is the consequence of a complex combinatory of bioactive molecules present in SEVs [173]. Not only proteins (cytokines, etc.) or lipids but also different RNA forms could account for these critical changes. In breast tissue, miRNAs regulate the expression of cytokines and growth factors that can affect extracellular matrix composition and pave the way for pathogenesis [174].

#### 4.2.1. Breast Cancer Cells-Derived Exosomes Transfer to Local Microenvironment

SEVs derived from BC cells can transform non-tumor breast ones. SEVs derived from MDA-MB-231 BC cells induced epithelial-mesenchymal transition (EMT) [175] while those derived from triple-negative breast cancers (TNBC) HCC1806 cell line induced proliferation and drug resistance in MCF-10 breast epithelial cells [176].

SEVs could transfer miR-370-3p from BC cells to normal fibroblasts, facilitating their activation through CYLD down-regulation and further NF-ζB signaling pathway [177], leading to cancer-associated fibroblasts (CAFs) with protumorigenic and proangiogenic properties [178] (Figure 3). miR-9 was also found to convert normal fibroblasts into CAFs, and its overexpression also identified a signature of different genes related to cell motility and extracellular matrix organization [179]. BC cell SEVs encapsulated miR-105 can mediate metabolic reprogramming of CAFs through Myc signaling [180].

TNBCs are highly infiltrated by tumor-associated macrophages (TAMs). TNBCs release SEVs and soluble molecules that promote, via TLR2 and TLR3 Toll-like receptors, monocyte differentiation toward TAM fates to phenocopy the tumor and rewire the microenvironment [181]. SEVs, combining either surface CSF-1 promoting survival or cargoes promoting cGAS/STING pathway, specifically promoted macrophage differentiation into proinflammatory TAMs bearing an interferon response signature [182]. Delivery of BC cell-derived SEVs containing miR-138-5p downregulates KDM6B expression inhibiting M1 and stimulating M2 polarization [183]. Likewise, lncRNA BCRT1 secretion mediated by BC exosomes promoted M2 polarization, further accelerating BC progression [184]. This SEVs-induced pro-survival macrophage differentiation is driven through IL-6 receptor beta/glycoprotein 130/STAT3 signaling pathway [185].

Additionally, surrounding endothelial cells (EC) can be activated by BC cell-secreted SEVs. Exosomal Annexin II (AnxA2) transfer from BC cells has been shown to promote EC angiogenesis [186].

Lastly, BC cells can also interact with adipose tissue [187] through exosome transfer. Normal adipocytes are driven into cancer-associated adipocytes by tumor cells [188] through SEVs transfer of oncomiRs [189]. BC cells-derived EVs can also convert adipose tissue-derived MSCs to myofibroblasts [190].

#### 4.2.2. The Microenvironment Produces Exosomes That Could Transform Breast Cancer Cells

In response to BC cells, TME modifications induce SEVs-driven stromal cell response, resulting in tumor changes by further modifying BC cells [191]. This continuous dual SEVs-driven interplay between stromal cells and BC cells is central in tumor behavior as it may drive either tumor cell proliferation or migration. Among TME, CAFs, ECs, and infiltrating TAMs are likely to be the major cell types interacting with BC cells or within the TME through SEVs signaling.

CAFs are well known to play a pivotal role in controlling cancer cell invasion and metastasis, immune evasion, angiogenesis, and chemotherapy resistance [192]. CAF-derived exosomes carrying miR-181d-5p can promote proliferation, invasion, migration, and EMT and inhibit BC cell’s apoptosis by downregulating CDX2 and its downstream target HOXA5 [193]. BC cells’ endocytosis of CAFs SEVs miRs -21, -378e, and -143 increased their capacity to form spheres, stem cell and EMT markers expression, and anchorage-independent cell growth [194]. Transfer of CAFs p85α-deficient SEVs carrying the Wnt10b protein into BC cells induced EMT [195]. CAFs SEVs could also reprogram BC cell metabolism by modulating pyruvate kinase PKM2 expression through the enrichment of exosomal noncoding RNA [196]. Once metabolically reprogrammed, miR-105 transformed CAFs promote glutamine and glucose metabolism to feed adjacent BC cells [180]. Lastly, SEVs transfer results in CAFs activation through miR-146a/TXNIP axis to activate the Wnt pathway, which in turn enhances the invasion and metastasis of BC cells [197].

SEVs’ transfer from EC drives a cadherin switch in BC cells that favors further intimate contacts between EC and BC cells [198]. Blocking IL-3R-alpha suppresses EC SEV-induced angiogenesis stimulation by targeting the Wnt/β-Catenin pathway [199].

Once transformed, TAMs can also transfer SEVs to BC cells [200]. A recent report shows that TAMs SEVs-driven noncoding RNA molecules transfer will boost BC cell proliferation and direct their phenotype and metabolic changes to progression and metastasis [201].

Adipocytes are, by mass, the preponderant non-malignant cell type in BC TME. Adipocyte tissue (AT)-derived SEVs can also enhance growth, motility, and invasion, induce stem cell-like properties, and specific EMT features in estrogen receptor (ER)-positive and TNBC cells [202]. Among the AT SEVs activated signaling pathways in BC cells are Hippo [203], HIF-1α [202], ERK [204], Wnt/β-catenin [205], JAK/STAT3 [206], PI3K/AKT, and TGFbeta/Smad [207] (For review, [208]).

#### 4.2.3. Local Inflammation at the Tumor Site and Extracellular Vesicles

Chronic inflammation is likely to be an essential driver in triggering tumor progression and metastasis [209]. In such process installation, SEVs are likely to play an important role [210]. Triple-negative TNBC cells release SEVs and soluble molecules promoting specific monocyte differentiation toward proinflammatory macrophages bearing an interferon response signature [182]. BC tumor-derived SEVs can induce an M1 proinflammatory response in macrophages through the activation of NFκB, which stimulates the production of inflammatory cytokines including GCSF, IL-6, IL-8, IL-1β, CCL2, and TNF-α [211]. NF-κB is a significant regulator of inflammation, and constitutive activation of NFκB is often observed in BC cells and associated with an aggressive phenotype. This M1 activation of NFκB, but also p38 MAPK and STAT3 pathways, seems to be triggered by high-level annexin A2 containing SEVs [186].

### 4.3. Promotion of Tumor Expansion

Accumulated genetic and epigenetic changes often activate the expression of oncogenes while silencing tumor suppressors during carcinogenesis [212]. BC genomic instability leads to several protooncogene mutations affecting multiple signaling pathways [213]. Cell cycle pathways (gain of function mutations of cyclin E, cyclin D, and CDK2/4/6) are transformed in about 50% of all BC types [214]. MAPK signaling is greatly amplified in 80% ERBB2-positive BC [215]. PI3K pathway is altered in more than 60% of luminal A BC [214] while 90% TBNCs undergo p53 inactivation [216]. SEVs released by CXCR4-positive BC cells increase the oncogenic potential of tumor cells in mice [217]. Hypoxic BC cells produce a high amount of SEVs containing long non-coding lncRNAs SNHG1, which, when upregulated, acts as an oncogene [218]. Their transfer, targeting the miR-216b-5p/JAK2 axis, promotes growth in vivo by upregulation of the JAK2/STAT3 pathway [219]. BC gain of function p53-containing small SEVs convert surrounding tumor microenvironment fibroblasts to cancer-associated ones [220]. Interestingly, many oncogenes, especially MYC and AURKB, can regulate either SEVs’ biogenesis or release in BC cells [221].

### 4.4. Cancer Metabolism Reprogramming

Throughout the natural history of cancer, tumor cells should unveil high metabolic plasticity to adapt to continual changes within the tumor and surrounding environment [222]. Tumor cell proliferation must continuously adjust their metabolism to meet the highest nutrient capacity to fulfill enhanced biosynthetic and bioenergetics demands. In normal cells, glycolysis and mitochondrial oxidative phosphorylation (OXPHOS) cooperate to produce energy. In BC, as mitochondrial ATP production is generally impaired, tumor cells enhance glycolytic glucose consumption to get sufficient ATP, thus generating a high lactate content even in aerobic conditions (“Warburg effect”) [223,224]. While increasing evidence associates metabolic reprogramming to OXPHOS and subsequently enhanced glutaminolysis with the induction and maintenance of the epithelial–mesenchymal transition program [225], the use of mitochondrial metabolism in BC cells migration and invasion is still controversial [226]. Mammary tumor-initiating cells (MaTICs) seem more dependent on OXPHOS, producing less lactate [227]. Triple-negative TNBCs mostly rely on glycolysis [228], while reprogramming to OXPHOS is associated with a higher risk of recurrence and death [229]. High lactate production and secretion induce tumor microenvironment acidification promoting immune surveillance escape and metastasis [230].

Recently, it has been evidenced that BC tumor microenvironment metabolism can largely modulate cancer cell progression [231]. Cancer-associated fibroblasts (CAFs) can provide metabolites that will facilitate tumor cells’ ATP production. Lactate, exported through CAFs MCT4 lactate shuttle then uptaken through cancer cells MCT1 lactate transporter, could be used to fuel surrounding cancer cells, a process called “reverse Warburg effect” [232,233]. Adipocytes’ free fatty acid (FA) secretion followed by free FA CD36 uptake promotes BC cells progression [234]. Either FA synthesis and FA oxidation or glutamine and serine metabolisms all increase in tumor cells as lipids, amino acids, and nucleotides are strongly required for their multiplication [235,236]. BC cells and surrounding tumor microenvironment cells can shed SEVs that will modulate cancer cell metabolism and play a role in their proliferation. SEVs can contain metabolites and metabolism enzymes that can modulate cancer cells’ metabolism. For example, GLUT-1 glucose transporter was enriched in BC cells SEVs [237]. MDA-MB-231 BC cell line SEVs increase the peripheral blood mononuclear cells’ expression of GLUT1 and hexokinase HK2 genes, which are effective in the glycolysis pathway [238]. BC cell-secreted miR-122 reprograms glucose metabolism in the premetastatic niche to promote metastasis [239]. MDA-MB-231 BC cell-derived SEVs led to pyruvate kinase M2 (PKM2) phosphorylation in MCF7 cells that acquired a more aggressive phenotype, which resulted in increased aerobic glycolysis and cell proliferation [240]. Exosomal miR-105 from BC cells can alter the glucose metabolism of stromal cells and thus promote the growth of cancer cells under nutrient-deprived conditions [241]. In addition, miR-105 combined with miR-204 targets RAGC to regulate mTORC1 upon amino acid stimulation. Affected fibroblasts exhibit reduced mRNA translation and selective protein synthesis [242]. miR-144 containing SEVs downregulates the MAP3K8/ERK1/2/PPARγ axis, thus inducing beige/brown differentiation, while miR-126 remodels metabolism by disrupting IRS/Glut-4 signaling and activating the AMPK/autophagy pathway in resident adipocytes [189]. Exosomal miR-155 promotes lipolysis in adipocytes and facilitates an aggressive phenotype of BC-derived tumor cells [243].

In parallel, SEVs from tumor microenvironment cells can modulate BC cells’ metabolism. SNHG3 knockdown in CAF-secreted exosomes suppressed glycolysis metabolism and cell proliferation by the increase of miR-330-5p and decrease of pyruvate kinase PKM expression in tumor cells. SNHG3 functions as a miR-330-5p sponge to positively regulate PKM expression, inhibit OXPHOS, increase GLYC, and enhance BC cells’ proliferation [196]. Metabolic remodeling in cancer-associated adipocytes surrounding BC cells enhances tumor aggressiveness by promoting cancer cell survival and proliferation through SEV production [244]. As for CAFs, miR-105 can also activate MYC signaling in cancer-associated adipocytes to induce a metabolic program secreting energy-rich metabolites to fuel neighboring cancer cells [241].

### 4.5. Angiogenesis Induction

Angiogenesis is an essential feature for tumor proliferation and further metastasis. The uptake of tumor-derived SEVs by ordinary endothelial cells activates angiogenic signaling pathways in endothelial cells (ECs) and stimulates new vessel formation [245]. Once internalized, SEVs are immediately directed to the perinuclear zone and actin filaments-rich area. When tubules are formed, SEVs move to the cell periphery and enter advanced pseudopods [246]. After complete remodeling, adjacent ECs probably transport SEVs to neighboring ECs and other cells within the tumor microenvironment [111]. In hypoxic conditions, BC cells can secrete angiogenic factors, such as VEGF-A, inducing ECs migration and tumor angiogenesis [247]. Aside from Notch signaling [248] and angiopoietins [249], VEGF is one of the more potent angiogenesis promoters, thus behaving as an important mitogen with high specificity for ECs [250]. Studies have reported that SEVs released from hypoxic tumors are more likely to cause angiogenesis and vascular leakage; hypoxia gradually promoting, through HIF-1α signaling, BC cells SEVs release [251]. SEVs transfer with miR-210 from hypoxic BC cells to cells into the tumor microenvironment-induced expression of vascular remodeling-related genes, such as Ephrin A3 and PTP1B, to promote angiogenesis [252]. The same miR-210 and a set of other angiogenic miRNAs are enriched in SEVs released by metastatic BC cells, a secretory process regulated by neutral sphingomyelinase 2 (nSMase2, SMPD2). These SEVs, once transferred to ECs, enhance the capillary formation and migration capability [253]. Docosahexaenoic acid (DHA) has potent anticancer properties, mainly through VEGF suppression [254]. A recent report showed that DHA increased the expression of anti-angiogenic miRNAs (i.e., miR-34a, miR-125b, miR221, and miR-222) while decreased levels of proangiogenic miRNAs (i.e., miR-9, miR-17-5p, miR-19a, miR-126, miR-130a, miR-132, miR-296, and miR-378) in SEVs derived from DHA-treated BC cells [255]. SNHG1 enclosed in BC cells SEVs induces angiogenesis via regulating the miR-216b-5p/JAK2 axis [219]. circHIPK3 enhanced MTDH expression in the EC by sponging miR-124-3p, favoring endothelial tube formation [256]. Another miR, miR-22-3p, mediated tumor vessel abnormalization by suppressing transgelin, thus promoting tumor budding and BC progression in vivo [257].

Aside from the various forms of SEV RNAs involved in its promotion, SEV-specific proteins can also stimulate angiogenesis. STIM1 promotes angiogenesis by reducing exosomal miR-145 in BC MDA-MB-231 cells [258]. Annexin II (AnxA2), a Ca2+-dependent phospholipid-binding protein associated with the plasma membrane, is one of the most expressed proteins in SEVs [259]. BC-derived SEVs transfer proangiogenic AnxA2 to ECs and induce angiogenesis by the tPA-dependent increase in plasmin generation [186]. Ephrin-A2 (EPHA2) was also rich in highly metastatic BC-derived exosomes and confers a proangiogenic effect [260]. Heparanase helps drive SEVs secretion and alters exosome composition (increase in matrix metalloproteinase-9 (MMP-9), VEGF, hepatic growth factor 2 (HGF2), and receptor activator of nuclear factor κ-B ligand (RANKL)) that impact both tumor and host cell behavior [261]. Lastly, EC-derived SEVs themselves can play a role in BC cells. ECs SEVs contained soluble and membrane-anchored forms of VE-cadherin that drive a cadherin switch in BC cells and neo-expression of VE-cadherin [198]. On the other hand, it has been previously reported that mesenchymal stromal cells MSC-derived SEVs negatively modulate angiogenesis by down-regulating BC cells’ VEGF synthesis through miR-16 transfer [262]. Additionally, MSC-derived SEVs enrichment with miR-100 suppresses angiogenesis in vitro by VEGF down-regulation through mTOR/HIF-1α)/VEGF signaling axis modulation [263]). Such findings emphasize SEVs’ multifaceted role in tumor-to-stroma communication within the TME.

### 4.6. Immune Evasion

Before the clinical presentation, most malignant cells are eliminated by immune surveillance through combined stimulation of innate and adaptive immune responses [264]. Nevertheless, BC cells, like other cancer cells, must evade immune control, a prerequisite in the transition from preinvasive to potentially lethally invasive disease [265]. In the early steps of tumor development, host immune factors play a crucial role in rejecting cancer cells [266]. Thus, clonal evolution patterns during progression will depend on the immune context [267]. Some progressing clones become immune privileged, despite tumor-infiltrating lymphocytes, while immunoedited tumor clones are eliminated [268]. To evade the immune system, tumors release immunosuppressive cytokines (e.g., TGF-β, interleukins IL8, IL6, IL10, etc.) and skew the tumor microenvironment to a more immunosuppressive one through either inhibiting CD8+ T cells, NK cells, dendritic cell maturation or increasing Tregs and tumor-associated macrophages (TAMs) [269]. SEV signaling in BC has been shown to play a crucial role in the crosstalk between immune and cancer cells [270]. Tumor-derived SEVs interacting with immune cells deliver negative signals to these cells and interfere with their antitumor functions [271,272].

The immunosuppressive nature of BC SEVs was confirmed in vitro where it promoted T-cell exhaustion and NK-cell cytotoxicity [273] (Figure 3). Hypoxia enhances SEVs secretion by BC cells, which acts to suppress T cell proliferation via TGF-β [274]. In the meantime, tumor-derived SEVs also promote Treg expansion and increase their immunosuppressive functions [275]. SEVs from 4T1 murine BC cells blocked the differentiation of myeloid precursor cells into CD11c+ DCs and induced cell apoptosis. They also drastically decreased CD4+IFN-γ+ Th1 differentiation but increased the rates of Treg cells [276]. While promoting the in vitro expansion of CD4(+)CD25(+)FOXP3(+) Treg cells and enhancing their suppressor activity, BC cells Mage 3/6 positive SEVs also inhibited signaling and proliferation of activated CD8(+) but not CD4(+) T cells and induced apoptosis of CD8(+) T cells [275]. BC cells SEVs SNHG16 lncRNA induced CD73 + γδT1 cells to act as immunosuppressive regulatory T cells by activating the TGF-β1/SMAD5 pathway [277].

Human MDA-MB-231–derived SEVs induce M2-type macrophage polarization (upregulation of CD206 and arginase-1), supporting enhanced tumor growth and axillary lymph node metastasis in an orthotopic triple-negative TNBC model [278]. miR-34a in triple-negative TNBCs mediate M1 polarization, while antagomiR-34a promotes M2 plasticity [279]. miR-503 in BC patients is vital in promoting brain metastasis by programming the microglia through M1 to M2 macrophage polarization induction [280]. Additionally, BC cells’ SEVs LncRNAs repertoire correlates with macrophage polarization [281]. Specifically, SEVs lncRNA BCRT1 promoted M2 polarization of macrophages, further accelerating BC progression [184]. The combination in TNBC SEVs of surface CSF-1 promoting survival and cargoes promoting cGAS/STING or other activation pathways led to the differentiation of this particular macrophage subset [182].

Cancer-associated fibroblasts (CAFs)-derived exosomes suppress immune cell function in BC by regulating PD-L1 levels in BC cells via the miR-92/LATS2/YAP1 pathway [282]. Lin28B promotes lung metastasis of BC cells by building an immune-suppressive pre-metastatic niche. Lin28B enables neutrophil recruitment and N2 conversion. The N2 neutrophils are then essential for immune suppression in the pre-metastatic lung by PD-L2 up-regulation and a dysregulated cytokine milieu [283]. Lastly, tumor-derived SEVs inhibited NK cell immunity using murine mammary (TS/A) tumor cell lines. TS/A SEVs are taken up by NK cells and account for decreased cytotoxic activity. Not only TS/A SEVs but also SEVs from human MDA231 or murine 4T1 BC cells could significantly block the proliferation of NK cells induced by IL-2 [284].

### 4.7. Metastatic Spread Induction and Secondary Settlement

Metastasis, which causes over 90% of BC-related deaths, behaves as a cascade comprising local invasion, intravasation, survival in the circulation, premetastatic niche modeling, and extravasation, and then metastatic niche colonization [285].

#### 4.7.1. Extracellular Vesicles and Epithelial to Mesenchymal Transition of BC Cells

Within the primary tumor, epithelial-to-mesenchymal transition (EMT) that confers enhanced mobile capabilities to tumor cells is likely to be one of the primary metastatic events. In BC, EMT activation has been shown to increase stemness [286], with most of the hematogenous circulating cancer cells harboring a mesenchymal phenotype [287]. Tumor-derived SEVs can facilitate EMT [288,289,290]. SEVs derived from mesenchymal stromal cells contained several molecules able to induce EMT such as well-known inducing proteins such as transforming growth factor-beta (TGF-β), hypoxia-inducible factor-alpha (HIF1α), or β-catenin as well as miRNAs, lncRNAs, and circRNAs [291]. CAF-derived exosomal miR-181d-5p can regulate CDX2 and HOXA5 in BC cells, thereby promoting their EMT [193]. Increased concentrations of MiR-9, miR-424, and miR-155 in SEVs led to BC cell EMT and aggressiveness [243,292,293]. SEVs are associated with pro-metastatic phenotype reprogramming in recipient surrounding cancer cells [294]. SEVs from human adipose-derived mesenchymal SCs promote migration through the Wnt signaling pathway in a BC cell model [205].

#### 4.7.2. Extracellular Vesicles Impact on Extracellular Matrix Disruption

The extracellular matrix (ECM) is also an essential regulator of BC progression [295]. BC SEVs can increase cancer cell invasion by containing mediators of cancer progression and critical factors in tissue remodeling, a prerequisite for seeding [296,297]. In that process, ECM stiffening due to excess deposition and crosslinking of collagen dramatically influences tumor behavior and fate by orienting fibers, thus facilitating metastatic cell intravasation [298]. Stiff ECM promotes SEV secretion in a YAP/TAZ pathway-dependent manner and triggers BC invasiveness using thrombospondin-1 (THBS1) as a master player [299]. BC cell-derived SEVs can also cargo ECM degradation enzymes such as MMPs, etc., as well as their regulators [290,300]. Tumor-derived SEVs transferred surface-bound proteases such as glycosidases to cleave ECM components, resulting in ECM remodeling and facilitating tumor development [296,301]. Interestingly, silencing Rab GTPases that tune biogenesis and secretion of pro-metastatic SEVs in BC cells, upregulate the levels of MCAM and CD146 adhesion molecules and limit BC metastasis [302].

To initiate the metastatic process, BC cells will also recruit and educate stromal cells to induce cancer-associated fibroblasts (CAFs), tumor-associated macrophages (TAMs) with the immune-suppressive M2 phenotype, and endothelial cells that promote tumor angiogenesis [303]. SEVs are likely to be critical players in this mandatory recruitment that contributes to the ability of BC cells to metastasize [304,305]. CAF-derived SEVs can promote BC cell motility through two independent mechanisms involving Wnt [306] or Notch [307] signaling in the cancer cells. Tumor SEVs mediate the migration of MDSCs and contribute to the metastasis of murine BC cells (4T1 cells) to the lung in a CCL2-dependent manner [308].

#### 4.7.3. Extracellular Vesicles and BC Cells Spread

Once the extracellular matrix is disrupted, the distant spread can then arise in two steps. The first concerns local tumor cell dissemination, where epithelial cells migrate through the tumor microenvironment at the front of the tumor through the generation of membrane protrusions (invadopodia) and basal lamina break-in [309]. It was clearly shown for BC using an injection of MTLn3 cells. This highly invasive rat mammary adenocarcinoma cell line forms invadopodia in vitro into the mammary gland of immunocompromised mice and rats and allows them to form tumors [310]. Cancer-associated fibroblasts CAFs-SEVs enhance this BC cell protrusive activity and motility via Wnt-planar cell polarity signaling [306]. The second involves vascular disruption to allow tumor cells hematogenous spread. Tumor SEVs can increase vascular permeability to promote the extravasation of circulating tumor cells (CTCs). Both exosomal miR105 and miR-939 secreted by metastatic BC cells are involved in VE barrier destruction, thereby increasing vascular permeability and promoting distant metastasis [180,311]. Identically, metastatic BC cells facilitate brain metastasis by releasing miR-181c-containing SEVs capable of destroying the blood-brain barrier [312].

#### 4.7.4. Extracellular Vesicles, Pre-Metastatic Niche, and Secondary Organ Settlement

As SEVs are not limited to the local tumor microenvironment, they can also cargo “tumor-nourishing” environments at distant sites to encourage metastatic settlement. SEVs educate a metastatic microenvironment, commonly defined as the pre-metastatic niche allowing circulating tumor cells (CTCs) to find a suitable environment in which they can settle and then proliferate. Such niche generation is characterized by local tissue inflammation, immune suppression, stromal cell activation, and ECM remodeling. Pre-metastatic niches are characterized by key tissue architecture, composition, and metabolism modifications, facilitating CTCs’ arrival, survival, and further expansion [313]. SEV-mediated intercellular interactions can generate a pro-metastatic tumor microenvironment [314]. By modifying glucose utilization by recipient pre-metastatic niche cells, BC-derived extracellular miR-122 can reprogram systemic energy metabolism to facilitate metastatic progression [239]. MiR-940 overexpression induced in MDA-MB-231 BC cells has been shown to induce extensive osteoblastic lesions in mice by facilitating the osteogenic differentiation of host mesenchymal cells [315].

An important characteristic of tumor cells relies on their capacity to colonize preferentially specific organs (organotropic metastasis) that are often determined by anatomic aspects. According to the Paget “seed and soil” theory [316], SEVs can even be considered as the “soil conditioner in BC metastasis” [317,318], leading to an inflammatory and mechanical niche promoting survival and colonization of immigrant CTCs. Indeed, bone marrow lesions were observed in mice bearing mammary cancer far before the arrival of tumor cells [319]. Integrin (ITG) SEVs repertoire seems to drive organ-specific metastasis, ITG α6β4 and αvβ3 on the surface of BV SEVs increasing lung metastasis [320]. During the establishment of an inflammatory environment in organs to which tumors will metastasize, SEVs contribute to the upregulation of proinflammatory cytokines and inflammation-activating factors, as well as the recruitment of immune cells to the pre-metastatic niche. BC-derived SEVs containing CCL3, CCL27, and other molecules are found to remodel the bone microenvironment, characterized by stimulating osteoclastogenesis and angiogenesis [321]. BC and lung tumor-derived SEVs containing Cell Migration-Inducing and hyaluronan-binding Protein (CEMIP) could induce a proinflammatory vascular niche by upregulating cytokines Ptgs2, TNF, and CCL/CXCL cytokines to promote brain metastasis [322]. Annexin A2 released by BC cells’ SEVs can induce macrophage-mediated activation of either p38 MAPK, nuclear factor κB (NF-κB), or STAT3 pathways, thus increasing IL-6 and tumor necrosis factor (TNF)-α secretion, thereby contributing to the formation of a premetastatic inflammatory microenvironment in distant organs such as the lung and brain [186]. Interestingly, once in the brain, BC cells’ survival is increased by SEV-encapsulated miR-19a released by astrocytes that act by decreasing PTEN expression [323].

Cancer cell SEVs can reprogram resident cells like in primary tumors to promote metastatic niche achievement and attract newly released CTCs. BC-derived exosomal microRNA-200b-3p uptaken by alveolar epithelial type II cells (AEC II) induces the high expression of CCL2, S100A8/9, MMP9, and CSF-1 in the lung to recruit myeloid-derived suppressor cells (MDSCs) and promote inflammatory pre-metastatic niche formation [324]. SEVs secreted by highly metastatic murine BC cells inhibit antitumor immune responses in premetastatic organs, directly suppressing T-cell proliferation and NK cell cytotoxicity [325]. BC cells SEVs remodeled lung parenchyma via a macrophage-dependent pathway to create an altered inflammatory and mechanical response to tumor cell invasion [326]. Immune cells can also play an important role in these distant organs [327]. BC-derived SEVs containing ANXA6 are targeted to the lung and activate the CCL2-CCR signaling axis, thus recruiting monocytes, which then differentiate into macrophages at this future site of metastasis [328]. These metastasis-associated macrophages (MAMs) have been first described in mouse models of BC lung metastasis [329]. At both bone and lung sites, MAMs promote BC cell extravasation, seeding, and metastatic outgrowth [330,331].

### 4.8. Cancer Cells Dormancy

Metastatic disease can occur years or even decades after the first diagnosis and subsequent treatment, suggesting that cells initiating recurrence are often long-lived and able to reactivate proliferation after long latency periods (also referred to as clinical dormancy) [332]. In BC, late recurrences (>5 years) account for most of the deaths among patients [333]. It is likely that these specific metastatic tumor cells exit the cell cycle and remain in a growth-arrested state [334]. Dormant tumor cells are commonly referred to as slow-cycling cancer stem cells that combine quiescent properties with tumor-initiating and chemoresistant properties, which favor later relapse and for the formation of metastases [335,336]. Dormant BC cells exhibit a distinct gene expression signature from metastatic ones regardless of the metastatic site [337]. When circulating tumor cells (CTCs) first extravasate from the vessels, they may reside in a niche surrounding the microvasculature, the perivascular niche (PVN) [338] that comprises resident hematopoietic cells, endothelial cells, and mesenchymal stromal cells (MSCs). Evidence has accumulated over recent years that the PVN in BC orchestrates CTC dormancy, principally responsible for cell survival and growth arrest [339]. CTCs may receive intrinsic factors (microenvironmental factors and signaling molecules) relevant to dormancy [340]. In bone marrow representing a niche for BC cell homing, SEVs from surrounding MSCs contain specific miRNAs that drive metastatic BC cells to dormancy. Gap junction-mediated import of microRNAs, including miR-222/223, mir-127, and mir-197 from bone marrow stromal cells, have been shown to elicit cell cycle quiescence in BC cells [341]. Interestingly, stromal SEVs are likely to also play an essential role as those containing miRNAs, such as miR-222/223, promote quiescence in a subset of BC cells [171]. Additionally, they also contribute to the dormancy of BC cells by reducing either CXCL12 levels [342] or targeting ERK1/2 signaling via miR-148a-3p [343]. Overexpression of MSCs-SEVs miR-23b in highly metastatic BC BM2 cells induced dormant phenotypes through the suppression of a target gene, MARCKS, which encodes a protein that promotes cell cycling and motility [344]. SEVs-enclosed miR-205 and miR-31, targeting the ubiquitin-conjugating enzyme E2N (UBE2N/Ubc13) and downregulating its activity, induced dormancy in MDA-MB-231 cells [345]. Interestingly, BC cells primed with MSCs SEVs were more highly resistant to chemotherapy [346]. In addition, SEVs from differentially activated macrophages influence the dormancy or resurgence of BC cells within the bone marrow stroma. M2 metastasis-associated macrophages (MAMs) form gap junctions with CSCs, resulting in cycling quiescence, reduced proliferation, and carboplatin resistance. Activation of M2 MAMs via the toll-like receptor 4 (TLR4) switched to the M1 phenotype can occur directly or indirectly through the activation of MSCs. Thus, M1 MAM SEVs activated NFκB reverse quiescent BC cells to cycling cells [347]).

### 4.9. Resistance to Therapy

BC is a heterogeneous disease in which each patient has individual characteristics that must drive treatment choices. In such a context, searching for new markers to improve the diagnosis and prognosis and achieve a better treatment response is mandatory. Currently, strategies for treating BC depend on the tumor subtype, and the selected treatments are directed to specific targets that are functionally altered in each cancer subtype. Conventional treatments for the management of BC patients have included endocrine therapy, targeted immunotherapy, and chemotherapy, all of these agents being used in adjuvant, neoadjuvant, and metastatic settings [5]. However, despite the improvement and diversification of therapeutics for BC patients and the emergence of new drugs during the last years, resistance to treatment remains a deadlock for women with an advanced BC for whom medicines no longer work.

#### 4.9.1. Resistance to Hormone Therapy

Nearly 80% of BC are estrogen receptor positive (ER+) [348], the vast majority of them being initially dependent upon the activation of ER by estrogens [349]. Because of the importance of the estrogen-ER axis in breast tumorigenesis, the main treatment options for these patients are still endocrine therapies such as aromatase inhibitors, selective modulators of ER activity, or selective ER down-regulators. Among all patients with BC who have hormone receptor-positive tumors, 84% receive hormonal therapy [350]. Nevertheless, the major challenge in treating ER+ BC is to overcome endocrine resistance, whose mechanisms can be very complex [351,352,353,354]. SEV secretion can be involved in these processes [355]. SEVs from tamoxifen-resistant MCF7 (MCF7TR) BC cells transfer resistance inducing epithelial–mesenchymal transition and resistance to apoptosis to wild-type ones [356]. Several SEVs transfers of noncoding RNAs conferring tamoxifen resistance have been reported: miR-221/222 that confers stem cell-like properties [357], miR-9-5p [358], circ_UBE2D2 by binding to miR-200a-3p [359], miR-22 [360], and UCA1 LncRNAs [361]. SEV-mediated transfer of mitochondrial DNA (mtDNA) has been shown to promote an escape of BC cells from metabolic quiescence and led to hormone therapy resistance both in vitro and in vivo [362]. Aside from tamoxifen treatment, enhanced SEVs production has also been reported in aromatase inhibitor-resistant BC cells [363].

#### 4.9.2. Resistance to Chemotherapy

Chemotherapy is generally used as neoadjuvant or adjuvant treatment in stage I-III BC (Miller 2022), while 60% of women diagnosed with metastatic disease (stage IV) most often receive radiation and chemotherapy alone [364]. Anthracyclines and taxane derivatives are two agents commonly used in the treatment of BC, but the emergence of chemoresistance often limits their efficacy. SEVs can be involved in the development of resistance. BC cells could export the chemotherapeutic drug doxorubicin (DOX) into the extracellular medium through vesicle formation, limiting its action [365]. SEVs isolated from the HCC1806 triple-negative TNBC cells can induce proliferation and drug resistance in the non-tumorigenic MCF10A breast cells [176].

Many studies have shown that resistance can arise through SEVs-mediated horizontal transfer of membrane-embedded drug efflux pumps to sensitive cancer cells. SEV-like structures containing the ATP-binding cassette (ABC) transporter protein ABCG2 have been reported to be increased in a variant of MCF-7 cell line with 20-fold resistance to mitoxantrone [366]. SEV delivery of P-gp (ABCB1) transporter was suggested to play an essential role in transferring drug resistance from DOX-resistant cells to drug-sensitive BC ones [367]. The transient receptor potential channel (TrpC) seems to play a crucial role in the upregulation of P-gp in drug-resistant BC cells [368]. Interestingly, SEVs from DOX-resistant MCF-7 cells were found to transfer TrpC5 (and also P-gp) to recipient human microvessel endothelial cells (HMECs) and further induce de novo expression of P-gp in these cells [369]. In sensitive MCF-7 cells, TrpC5-containing SEVs internalization led to Ca2+ influx through TrpC5s, which resulted in the upregulation of P-gp [370]. Interestingly, a strong correlation in nonresponsive tumors was observed between treatment resistance and increased TrpC5 expression by immunohistochemistry on BC patients’ tissues [371]. Ubiquitin carboxyl-terminal hydrolase-L1 UCHL1 (UCH-L1) was also found to upregulate P-gp expression by activating the MAPK/ERK pathway. UCH-L1-containing SEVs secreted by DOX-resistant human BC cells were taken up by DOX-sensitive human BC cells in a time-dependent manner and ultimately contributed to the chemoresistance phenotype [372]. SEVs transfer can confer neither DOX nor docetaxel (DTX) nor paclitaxel (PTX) resistance. PTX treatment induced the secretion of survivin-enriched SEVs from MDA-MB-231 cells, which highly promoted the survival of PTX-treated fibroblasts and SK-BR-3 cells [373].

Aside from protein transfer, noncoding RNAs have been involved in chemotherapy resistance induction. Resistant cells SEVs miR-100, miR-222, and miR-30a transfer confer resistance to wild-type ones [374]. Transfer of miR155 by inhibiting tetraspan 5 and promoting stemness [375,376], miR1246 by inhibiting Cyclin-G2 induced BC cells resistance [377], and miR-887-3p by targeting BTBD7 and activating the Notch1/Hes1 signaling pathway [378] can induce chemoresistance. Adaptation of cancer stem-like cell traits through SEVs transfer, including miR-9-5p, miR-195-5p, and miR-203a-3p targeting ONECUT2 has been shown to confer resistance [172]. SEVs derived from cisplatin-resistant MDA-MB-231 cells are characterized by a high expression of miR-423-5p that can be transferred to non-resistant cells [379]. LncRNA H19 was strikingly overexpressed in DOX-resistant BC cells and encapsulated into SEVs to transfer drug resistance. Similarly, the downregulation of H19 reversed DOX chemoresistance in sensitive BC cells [380].

Up to date, to overcome chemoresistance in BC treatment, several large-scale validation studies have been performed to determine the exosomal protein and miRNA expression profiles in drug-resistant BC after chemotherapy to find new potential markers and to better understand the transmission of SEVs-mediated chemoresistance [381,382,383,384,385].

#### 4.9.3. Resistance to Radiotherapy

Resistance transfer through SEVs seems to also be a potent mechanism that confers resistance to radiotherapy [386]. SEVs derived from radioresistant cells can increase cell viability and colony formation in naïve recipient ones and increase their radiotherapy resistance [387]. Cargo from irradiated cell-derived SEVs was distinct from non-irradiated cells, indicating alterations in the exosomal formation system [388]. SEV levels of proteins such as PERP, GNAS2, GNA13, ITB1, and RAB10 correlate with BC cell trastuzumab response [389]. X-ray irradiation activates SEV biogenesis and secretion in a dose-dependent manner in MCF7 cells and induces their resistance to radiotherapy [390]. Such effects of radiation not only concern BC cells but also surrounding tumor microenvironment ones. In a mouse BC model, SEVs derived from irradiated cells elicited immune responses of tumor-specific CD8+ T cells and inhibition of tumor size [391]. Likewise, radioresistant SEVs stimulate tumor-supporting fibroblast activity, facilitating tumor survival and promoting cancer stem-like cell expansion [392].

#### 4.9.4. Resistance to Targeted Therapy

As human epidermal growth factor receptor 2 (HER2) overexpression is often associated with BC poor prognosis, HER2-targeted therapy has been developed and achieved excellent efficacy in treating HER2+ BC [393]. Whereas trastuzumab generally has an excellent initial clinical response, most BC patients turn refractory to HER2-targeted drugs as early as one year after initiation of treatment. SEV-containing lncRNA AFAP1-AS1 (AFAP1 antisense RNA 1) overexpression was associated with poor prognosis in triple-negative TNBC patients where its upregulation activated Wnt/β-catenin pathway to promote tumorigenesis and cell invasion by increasing the expression of C-myc and epithelial-mesenchymal transition-related molecules [394]. It also functioned as a miR-2110 sponge to increase Sp1 expression, the AFAP1-AS1/miR-2110/Sp1 axis behaving as a potent modulator of the proliferation, migration, and invasion of triple-negative TNBC cells [395]. Another lncRNA, AGAP2-AS1, is also dysregulated in trastuzumab-resistant BC cells and plays a critical role in enhancing trastuzumab resistance by packaging into SEVs in an hnRNPA2B1-dependent manner [396]. SEV-mediated transfer of circHIPK3 also enhanced trastuzumab resistance [397]. SEV miR-1246 and miR-155 presence can be used as predictive and prognostic biomarkers for trastuzumab-based therapy resistance in HER2-positive BC [398]. Aside from the direct effect on cell proliferation, SEVs can enhance resistance to the anti-tumor immune response. SEVs from HER2-resistant cells have increased amounts of the immunosuppressive cytokine TGFβ1 and the lymphocyte activation inhibitor PD-L1, suggesting that they can induce immune evasion through neuromedin U [399].

CDK4/6 inhibition is now part of the array of targeted tools for patients with ER+ BC. SEVs’ miR-432-5p levels were higher in CDK4/6 resistant patients. Increased CDK6 expression is commonly observed in resistant cells and depends on TGF-b pathway suppression via miR-432-5p expression [400]. High baseline CDK4 mRNA levels in SEVs have been associated with response to palbociclib plus hormonal therapy, while the increase in TK1 and CDK9 mRNA copies/mL is associated with clinical resistance [401]. Deep proteomic analysis of plasma SEVs from resistant patients will help better understand underlying resistance mechanisms and give new potential resistance biomarkers [402].

To the best of our knowledge, no reports involving SEVs in the resistance process of BC cells have been yet reported for either other kinases, PARP, PI3K, mTOR, or immune checkpoint inhibitors.

## 5. Exosomes as Relevant Breast Cancer Biological Markers

Diagnosing BC in the early stages can make an essential difference in the patient’s treatment and prognosis. Aside from breast imaging which is crucial in the screening, diagnosis, and preoperative work-up of BC, biomarkers can provide additional insight into a patient’s diagnosis, prognosis, and response to treatment. Numerous biomarkers are currently used in BC management, notably tissue marker expression of different receptors (estrogen receptor (ER), progesterone receptor (PR), and human epidermal growth factor receptor 2 (HER2)) that is daily used for patients’ staging [403,404]. Aside from tissue ones, blood biomarkers are attractive means to monitor disease recurrence or progression, to follow treatment response, or to evidence targetable mutations that will direct therapy. Nevertheless, while iterative measurements of serum proteins such as CA15-3, CA27-29, and CEA have proved valuable tools in advanced cancers to monitor BC response to treatment, their poor sensitivity in early BC impairs their use in either diagnosis or prognosis [405]. New predictive and prognostic protein markers are still mandatory [406]. Liquid biopsies, which are supposed to get tumor-derived materials such as tumor DNA, RNA, and intact tumor cells in body fluids, are less invasive than tissue biopsies. They appear as an alternative for discovering new biomarkers for BC screening to diagnosis, prognosis, treatment response, and discovery of relapse [407,408]. As part of liquid biopsy, SEVs can be detected in patients’ biological fluids, such as blood, urine, CSF, and saliva [38] where they remain stable and protected from the degradation of serum ribonucleases and DNases [409]. SEVs can now be easily isolated [410] even though a universal standardized and widely accepted method for isolating and then analyzing SEVs is still mandatory [411,412,413,414]. As several miRNAs, lncRNAs, and proteins are differentially expressed in SEVs originating from tumor and normal cells, they are likely to be potential sources of biomarkers and become a promising field in BC management (Figure 4).

As valuable BC biomarker sources, SEVs can be divided into surface protein biomarkers and intraluminal biomarkers (mostly nucleic acids, among which figure miRNAs).

### 5.1. Exosome Nucleic Acid Cargo as Biomarkers

#### 5.1.1. Exosome mRNAs as Interesting BC Markers

Messenger RNAs (mRNAs) encapsulated within SEVs are transferred to recipient cells and translated into proteins, altering the behavior of the cells [166,415,416]. Highly cancerous cells communicated with less cancerous cells through SEVs transfer, increasing migratory behavior and metastatic capacity [417]. Fragile mRNAs encapsulated within the SEVs’ phospholipid bilayer structure are protected from the harsh external environment, which would otherwise degrade them [81,418]. While SEVs transcriptomic profile reflects only partly that of the cell of origin [419], the whole transcriptomic analysis identifies a global SEV mRNA signature and BC signal in patients [420]. A typical “stemness and metastatic” signature was reported in SEVs of patients with worse prognosis. This signature comprises several mRNAs, such as those coding for NANOG, NEUROD1, HTR7, KISS1R, and HOXC6 [217].

#### 5.1.2. SEVs miRNAs as Relevant BC Biological Markers

Circulating miRNAs (c-miRNAs) can travel in the bloodstream in two forms: in cell-free miRNAs (Ago2-related) or embedded in circulating tumor cells (CTCs), apoptotic bodies, or SEVs [421]. SEVs miRNAs are stable as they are protected from serum RNases [422]. Many papers have been published about using c-miRNAs in BC (for review [423]). A vast number of those report their modified expression either being up-or downregulated (for extensive reviews, [424,425,426,427]).

Either single miRNA (miR-21 [428], miR-155 [429], miR-223-3p [430], mir-373 [431], and mir-7641 [432]) or a combination of two or multiple miRNAs have been reported. Indeed, numerous reports have designed specific miRNA sets associated with BC. Combination of two plasma SEVs miRNAs (miR-21 and miR-1246 [433], miR-21 and miR-221 [434], miR-21 and miR-155 [435], and miR-92a, and miR-25-3p [436]), three miRNAs (miR-16, miR-30b, and miR-93 [437]; miR-21, miR-105, and miR-222 [438]; miR-21, miR-155, and miR-365 [439]; and miR-145, miR-155, and miR-382 [440]), four miRNAs (miR-21, miR-55, miR-10b, and Let-7a [441]), up to thirteen miRNAs (miR-21-3p, miR-192-5p, miR-221-3p, miR-451a, miR-574-5p, miR-1273g-3p, miR-152, miR-22-3p, miR-222-3p, miR-30a-5p, miR-30e-5p, miR-324-3p, and miR-382-5p) [442] have been described. To optimize and find new relevant miRNA combinations, some algorithms have been developed to detect BC specifically ([443].

As diagnosing BC at an early stage is still challenging, several sets of c-miRNAs have been specifically assayed for that purpose. Both sensitivity and specificity of SEVs miR-17-5p concentration were superior to conventional serum biomarkers such as CEA and CA15-3 [444]. A combination of five miRNA, miR-1246, miR-1307-3p, miR-4634, miR-6861-5p, and miR-6875-5p, was shown to detect BC with high sensitivity, specificity, and accuracy, even in the case of ductal carcinoma in situ (DCIS) [445]. Both increased concentrations of miR-21-5p and miR-10b-5p levels in serum-derived SEVs of BC patients correlate with BC grade [446]. The overall expression of nine microRNAs was higher in patients with stages I, II, and III compared to stage IV, with potential utilization for early detection [447]. Serum miR-423-5p was significantly associated with the tumor’s clinical stage and Ki-67 level [448]. A combination of miR-375, miR-655-3p, miR-548b-5p, and miR-24-2-5p has been found relevant for early BC diagnosis [449]. A dual microRNA signature based on miR-30b-5p and miR-99a-5p levels in plasma is a good diagnostic biomarker for BC [450]. Combinations of four miRNAs (miR-1246, miR-206, miR-24, and miR-373) were reported to have a sensitivity of 98%, a specificity of 96%, and an accuracy of 97% for BC detection [451]. Very recently, the miR-15a, miR-16, and miR-221 combination turned out to be promising for BC diagnostic [452].

BC prognosis is also an important issue. A recent review reported that 110 aberrantly expressed miRNAs have been associated with prognosis in BC [453]. Association of miR-126, miR-122, miR-92-1, miR-19a, and miR-29c together with circular miRNAs, such as miR-21-5p, miR-96-5p, and miR-125b-5p, can provide a promising evaluation marker in BC prognosis [454]). Moreover, a specific set of plasma SEVs miRNAs can be used for staging to evidence BC subtypes. Analysis of SEVs derived from plasma of 435 HER2+ and TNBC subtypes has identified 18 exosomal miRNAs that differed between HER2+ and TNBC subtypes, nine miRNAs also differing from healthy women [455]. Association of miR-34 and miR-520 can be used for ER+ and TNBC subtypes [454]. Such kinds of sets can also be used to predict early recurrence or metastasis. Seven miRNAs were differentially expressed between BC patients with and without recurrences, including four miRNAs upregulated (miR-21-5p, miR-375, miR-205-5p, and miR-194-5p) and three miRNAs downregulated (miR-382-5p, miR-376c-3p, and miR-411-5p) [456]. The association of miR-19a, miR-20a, miR-126, and miR-155 can discriminate against the metastatic outcome of BC patients [457,458].

A set of dysregulated selected exosomal miRNAs that could modulate target genes responsible for MAPK, TGF-beta, Wnt, mTOR, and PI3K/Akt signaling pathways have been associated with DOX resistance [384]. A specific miRNA signature was differentially expressed in SEVs derived from adriamycin-resistant (A/exo) and parental breast cancer cells (S/exo), with 309 miRNAs being increased and 66 significantly decreased in A/exo compared with S/exo [383]. The association of miRNAs targeting metabolic pathways has been reported with differential response to neoadjuvant chemotherapy (NACT) [459]. Three miRNAs before NACT (miR-30b, miR-328, and miR-423) predicted complete pathological response (pCR) in BC while upregulation of miR-127 correlated with pCR in triple-negative TNBC patients [460]. In a meta-analysis review, 60 of 123 reported miRNAs in the literature were found to be related to NACT response [461]. Dynamic evaluation of three miRNAs, including miR-222, miR-20a, and miR-451 was associated with NACT chemo-sensitivity [462]. Interestingly, a combined signature of four miRNAs (miR-4448, miR-2392, miR-2467-3p, and miR-4800-3p) could be used to discriminate between chemotherapy responders and nonresponders TNBC patients [463].

Given the vast number of publications on miRNA differential expression in BC patients’ SEVs, the development of relevant meta-analysis is strongly mandatory, and several have so far been performed. One suggested that miR-21 is likely to be a potential biomarker for early diagnosis, with high sensitivity and specificity being significantly upregulated in BC [464]. This result was confirmed later [465]. Another reported that plasma SEVs miR-23b upregulation is linked to poor overall BC survival [466]. A third one pinpoints miR-9 as an interesting BC biomarker [467]. Globally, there is little consistency among the circulating miRNA signatures identified in these different studies, mainly due to the lack of standardization and result reproducibility, which remains the most significant issue [468]. So far, no panels of circulating miRNAs are still ready for BC diagnosis in a clinical setting [469,470].

#### 5.1.3. SEVs lncRNAs as Interesting Emerging Biomarkers

Long-noncoding RNAs are regulatory transcripts longer than 200 nucleotides that play an essential part in many fundamental cellular processes [471], and their deregulation is considered to contribute to carcinogenesis [472] and metastasis [473,474]. The increased serum concentration of several SEVs lncRNAs has been associated with poor prognosis in BC patients. LncRNA DANCR [475] and lncRNA metastasis-associated lung adenocarcinoma transcript 1 (MALAT1) [476] have been associated with BC worsened evolution. Serum exosomal lncRNA XIST has been described as a potential biomarker to diagnose TNBC recurrence [477]. Fifteen exosome-related differentially expressed lncRNAs were recently identified to be correlated with BC prognosis [478].

Overexpression of specific lncRNAs has been evidenced as a marker of treatment resistance. Trastuzumab resistance is associated with the action of LncRNA OIP5-AS1 through miR-381-3p/HMGB3 axis [479], lncRNA ATB by competitively binding miR-200c, upregulating ZEB1 and ZNF-217, then inducing EMT [480], lncRNA AGAP2-AS1 by inducing BC cells autophagy [481], and lncRNA SNHG14 [482]. Higher expression levels of exosomal lncRNA-H19 compared to parental cells have been reported in DOX resistance [380]. A recent meta-analysis has confirmed that lncRNAs in SEVs could be a promising bioindicator for the diagnosis and prognosis of solid tumors [483].

#### 5.1.4. SEVs Circular Nucleic Acids as New Potential Diagnostic Tool

Extrachromosomal circular DNA (circDNA) is a type of cell-free DNA (cfDNA) that is more structurally stable than linear cfDNA currently used for cancer-related detections in clinical settings [484]. CirDNA is resistant to the action of extracellular nucleases due to the formation of macromolecular complexes with proteins (including histones) [485]. Commonly observed in both standard and cancer cells (Wang 2021), it can bind to the outer surface of exosomes (Tamkovich 2016) (Tutanov 2022) and be detected in serum (Ling 2021).

Aside from circDNA, circular RNAs (circRNA) also exist. CircRNA is a class of covalently closed single-stranded circular RNA molecules without free 5′ or 3′ ends [486]. Unlike traditional linear RNAs such as lncRNAs and miRNAs, circRNAs were not degraded by RNases or RNA exonucleases and were more stable and conserved in peripheral blood or plasma [487]. CircRNAs can exert various functions according to their parental genes, among which figure their ability to serve as a sponge for multiple miRNAs, suppressing their activity [488]. Many circRNAs have been discovered in various cancers, and they are activated in either inhibiting tumor progression or promoting tumorigenesis.

Both cDNA and circRNA can be observed in BC and hold promise to be used as SEVs biomarkers. CirRNA circ_0004771 accelerates BC cell carcinogenic phenotypes via upregulating dimethylarginine dimethylaminohydrolase 1 (DDAH1) expression through absorbing miR-1253 [489]. Circ_0000615, which was spliced from the ZNF609 gene, displays an expression level markedly upregulated in BC cell lines compared with normal ductal epithelial cells. It displays a better diagnostic efficiency in BC patients than routine tumor biomarkers such as CA153, CA125, and CEA. Its high expression was closely associated with advanced tumor stage, lymph node metastasis, and high grade of recurrence risk [490]. In TNBC, several cirRNAs are likely to be exciting biomarkers. lncRNA MALAT1 (metastasis-associated lung adenocarcinoma transcript 1) regulates linear isoforms of VEGFA, inducing the back-splicing of VEGFA exon 7 and producing circular RNA circ_0076611. Circ_0076611 is detectable in TNBC cells [491]. The expression of circHSDL2, targeting let-7a-2-3p during the progression of TNBC, was found to be significantly upregulated in serum SEVs and tumor tissues from TNBC patients [492]. Overexpression of circ-proteasome 20S subunit alpha 1 (circ-PSMA1) promoted tumorigenesis, metastasis, and migration through miR-637/Akt1/β-catenin (cyclin D1) axis in TNBC both in vitro and in vivo. circ-PSMA1 is upregulated in vitro in TNBC cells’ SEVs and in SEVs isolated from triple-negative TNBC patients’ sera [493].

### 5.2. SEVs Protein Cargo as a Source of New Cancer Biomarkers

Proteins located on the surface of and within SEVs may also be used as relevant cancer biomarkers as they may differ between healthy individuals and BC patients [494]. SEVs surface protein markers such as members of the tetraspanin family (CD9, CD63, CD81, CD82, and CD151), some integrins, multivesicular bodies (MVBs) formation proteins (TSG101, Alix, and Clathrin), and lipid raft proteins (flotillins) [495]. The level of CD82 was significantly higher in the serum of BC patients compared to the healthy controls, while the expression of CD82 significantly increased with malignant BC progression [496]. On the contrary, CD151-deleted SEVs significantly decreased the migration and invasion of TNBC cells [497]. In addition to these self-proteins for constructing SEVs, several proteins from BC-derived cells are likely to be potential biomarkers for early screening and diagnosis.

Enzymes and specific signaling proteins (EpCAM, EFGR, and survivin-2B) along with metalloproteinase ADAM10, heat-shock protein HSP70, and Annexin-1 can also be evidenced as general marker proteins detected in serum and pleural effusion-derived SEVs from BC cell lines or BC patients [237,498]. Epithelial cell adhesion molecules such as EpCAM and CD24 could be used as markers to identify cancer-derived SEVs in ascites and pleural effusions from BC patients [498]. As compared to healthy controls, higher levels of SEVs with glypican 1 (GPC1) on their surface (GPC1+) are found in BC patients’ sera (Melo 2015). GPC-1, glucose transporter 1 (GLUT-1), and disintegrin ADAM10 were potential TNBC biomarkers [237]. FAK and EGFR proteins can also be found, where FAK presence in SEV fractions is associated with in situ and stages I–III, while EGFR is associated with in situ and stage I BC [499,500]. SEVs containing amphiregulin (AREG) which binds to cell surface EGFR, were revealed to increase receptor BC invasive ability of cells [501] and can be used as prognostic and/or predictive markers [502]. Comparative proteome analysis of circulating SEVs in healthy and BC patients has shown that the association of three favorable (Serpin A1 (SERPINA1), keratin 6 (KRT6B), and SOCS3) and one unfavorable (insulin growth factor 2 receptor (IGF2R)) SEV protein markers allow diagnosing with 73% sensitivity and 100% specificity BC stage I and II [495]. The diagnostic value of fibronectin [503] and developmental endothelial locus-1 (Del-1) [504] in BC cell-derived SEVs were reported to display a sensitivity of 94.70% and a specificity of 86.36%. Some other markers are promising. Serum SEV annexin2 (AnxA2) holds promise as a potential prognosticator of TNBC as it is high in African American women with TNBC (Chaudhary 2020). The distinct expression pattern of SEV survivin-2B in serum is considered a sign of early-stage BC [505]. Some specific SEVs proteins have been correlated to response to treatment. Enrichment of CD44 in SEVs of doxycycline DOX-treated BC cells promotes their chemoresistance [506]. Programmed death ligand-1 (PD-L1), an essential immune checkpoint molecule, is expressed on BC SEVs and correlated with the progression and immunotherapy response [507]. Plasma SEV NGF concentration in BC patients undergoing neoadjuvant chemotherapy is associated with significantly poorer overall survival [508].

## 6. SEVs as Attractive Targets to Inhibit BC

SEVs are a source of cancer dissemination and a promoter of patients’ resistance to treatment. It is, therefore, mandatory to explore new therapeutic possibilities to suppress SEV-induced tumor progression and reduce SEV-related drug resistance.

### 6.1. Inhibition of SEV Uptake by Target Cells

The first possibility to limit SEVs’ adverse effects would be to inhibit their uptake by target cells [509]. SEV uptake capability has been reported to vary depending on the recipient cell type but not on the donor cell type [109]. It largely depends on surface molecules and glycoproteins on the vesicle membrane and the plasma membrane of the recipient cell [510]. Multiple uptake mechanisms are involved in the cellular internalization of SEVs, including caveolin- or clathrin-dependent endocytosis, macropinocytosis, phagocytosis, lipid raft-mediated internalization, and membrane fusion [510,511]. Many studies have found pharmacological inhibitors that could inhibit SEVs internalization. Heparin can inhibit SEVs uptake in a dose-dependent manner through direct action on heparan sulfate proteoglycans which themselves play a role in SEVs endocytosis [512]. Both cytochalasin D, through a direct inhibitory effect on actin polymerization, and methyl-β-cyclodextrin (MβCD), by depleting membranes’ cholesterol hence disrupting lipid rafts stability, inhibit phagocytosis/endocytosis mechanisms, and thus SEV uptake [513,514]. Disruption of clathrin-mediated and caveolin-dependent endocytosis by chlorpromazine or dynasore, a specific inhibitor of dynamin 2, as well as macropinocytosis inhibition by amiloride or omeprazole (OME) also inhibits SEVs uptake [515,516,517,518]. However, the extensive repertoire of mechanisms involved in SEV uptake in cancer impairs the overall efficiency of these molecules.

### 6.2. Inhibition of SEV Biogenesis

Another way to limit SEVs action would be to inhibit SEV biogenesis. Such an issue involves complex mechanisms and is likely to be challenging to implement. However, many pharmacological agents have been found and seem promising. The fluidity of the cell plasma membrane is fundamental during membrane lipid bilayer re-organization and SEV formation. In cancer, lipid mediators such as sphingosine 1-phosphate and ceramide, which are known to be associated with inflammation [519], also regulate SEV production [32,520]. Sphingomyelinases, acid (SMPD1), and neutral sphingomyelinase (SMPD2) are ubiquitous enzymes required for ceramide synthesis that can be specifically inhibited. GW4869, cambinol, and spiroepoxide inhibit SMPD2. Blocking SMPD2 by either GW4869 drug or specific SMPD2 siRNA results in a dose-dependent inhibition of SEV release [521]. GW4869 blocks SEV biogenesis by preventing the ceramide-modulated inward budding of multivesicular bodies and the subsequent release of SEVs [522]. Complementarily, SMPD2 overexpression increases miRNAs’ extracellular amounts [523]. The link between SMPD2 and SEVs has been associated with BC aggressiveness [253]. The association of OME that inhibits SEVs uptake with GW4869 that limits SEV biogenesis reduces paclitaxel (PTX) amount in SEVs, thus increasing the therapeutic effect of PTX on BC cells [524]. As GW4869 seems promising, imipramine, a tricyclic antidepressant, is also a source of interest because of its inhibitory activity on SMPD1 [525,526].

TSG101 is a protein involved in endosome trafficking and SEV biogenesis [527]. TSG101 knockdown in BC cells induces apoptosis and inhibits proliferation, suggesting that TSG101 is a potential therapeutic target in cancer [528].

#### SEV Release Inhibition

A third possibility to target SEVs relies on limiting or inhibiting their release. A drug that can inhibit SEV release is manumycin A, an antibiotic that is a selective and robust inhibitor of Ras farnesyl transferases. Farnesyltransferase inhibitors inhibit Ras activity and, therefore, SEV release [522]. Rasal2, a Ras-GTPase-activating protein (RasGAP), is a known tumor suppressor in luminal B breast cancer, frequently metastatic and recurrent. Rasal2 knockout (KO) in MCF-7 cells enhanced SEVs release and increased autophagy-related proteins in exosomal fraction while attenuated by SEV release inhibitor GW4869 (Wang 2019). Aside from Ras proteins, there are also Rab proteins that are also modulators of SEVs biogenesis [12].

Interestingly, associated with an increase in SEV secretion, the most up-regulated proteins in long-term estrogen-deprived MCF-7 LTED cells were represented by Rab GTPases [363]. Among Rab proteins, Rab27a and Rab27b seem to play a significant role in SEVs docking and exocytosis [48] and are involved in mammary gland development [125] and cancer [529]. Either knockdown of Rab27a in lung cancer [530] or gold nanoparticles conjugated with anti-sense Rab27a oligonucleotides to mute Rab27a in BC [531] generate significant inhibition of SEV release.

As plasma membrane fluidity is essential for SEV shedding, drugs targeting lipid raft formation or cholesterol synthesis will interfere with SEV release. Lipid depletion results in SEV release reduction (Skotland 2017). Pantethine, a pantothenic acid (vitamin B5) derivative, is used as an intermediate in the production of coenzyme A and plays a role in lipid metabolism, reducing total cholesterol levels. Pantethine inhibits cholesterol synthesis by 80% and fatty acid synthesis (Ranganathan 1982). Pantethine prevents murine systemic sclerosis by inhibiting microparticle shedding (Kavian 2015), an effect also observed on chemoresistant BC cells (Roseblade 2015).

Actin and actin-regulating proteins are also strongly involved in SEV secretion. Invadopodia are cellular structures used by cancer cells to degrade extracellular matrix and invade. Because of high levels of actin, such structures are critical sites for EV release. Indeed, invadopodia inhibition limits EV release [532]. Targeting cortactin, the actin-nucleation-promoting factor acting as an actin dynamics regulator, decreased SEV release, whereas its overexpression increased [533]. The non-receptor tyrosine kinase Pyk2 is highly expressed in BC and mediates invadopodia formation and function via interaction with cortactin. Targeting Pyk2 with a specific Pyk2-derived peptide inhibits invadopodia-mediated breast cancer metastasis [534].

Other drugs targeting SEV release have been used in BC. A novel anti-cancer SMR peptide that antagonizes BC cell SEVs release results in cell cycle arrest and tumor growth suppression [535]. D-Rhamnose β-hederin (DRβ-H), a novel oleanane-type triterpenoid saponin [536], and shikonin, a naphthoquinone [537], both isolated from traditional Chinese medicinal plants, attenuate resistance traits in doxycycline DOX-resistant BC cells and reduce tumor burden by decreasing SEV secretion. Cannabidiol (CBD) has been reported to be a potential inhibitor for SEV release in BC as it inhibits, in a dose-dependent manner, SEV release in MDA-MB-231 cells [538]. PEG-SMRwt-Clu, a drug derived from the secretion region of HIV-1 Nef protein, regulates exosomal pathway trafficking and seems promising. PEG-SMRwt-Clu was able to inhibit cell growth in BC cell lines and, more interestingly, to increase chemosensitivity partially. PEG-SMRwt-Clu was also associated with a decrease in the number of released SEVs [539].

Despite the current efforts and the number of SEV endocytosis, biogenesis, and release inhibitors already available, SEV inhibition remains a very complex issue because of the multifactorial nature of the different pathways involved in these processes. Nevertheless, there is no doubt that SEV uptake, biogenesis, or release inhibition is still a potential and attractive therapeutic cancer target.

## 7. SEVs as Nanovectors to Drive Therapy in BC

SEVs are significant players in tumor progression via the transfer of the cargo within them. Another possible way to cure BC would be to use an SEV-based therapy that uses SEVs as therapeutic nanovectors.

In the very last years, several reports have mainly focused on the idea that SEVs could be natural delivery vehicles to transport therapeutic drugs, antibodies, or RNAs to modify gene expression, especially in the cancer field [540,541,542,543,544,545,546] with a specific dedication to BC [547,548,549,550]. Indeed, SEVs are biocompatible, biodegradable, and, therefore, less toxic and immunogenic than other nanoparticle drug delivery systems such as liposomes or polymeric nanoparticles [551]. SEVs have innate limited immunogenicity and cytotoxicity [552] and can pass through anatomical barriers [553]. Additionally, as SEVs avoid drug degradation by extracellular enzymes, drug stability is enhanced [554]. Altogether, SEV’s capacity to target tumor cells is ten times higher than liposomes of a similar size. Such property is undoubtedly linked to particular ligand-receptor interactions and to efficient endocytosis mechanisms linked to the SEV membrane lipid composition that contributes significantly to cellular adherence and internalization [555].

Several reports have demonstrated the potential of using SEV therapy, and clinical trials are currently underway to find the best treatments that extend patient survival. Many kinds of SEV-based therapies have been shown to improve chemotherapy effectiveness. SEVs have been used to deliver chemotherapeutic drugs such as paclitaxel (PTX) [556,557,558] or doxycycline (DOX) [559,560,561]. While loading DOX in SEVs reduces its cardiotoxicity [562,563], it also enhances its efficacy when compared to traditional administration [562,564]. Packaging DOX into SEVs increases its stability, thus allowing a better collection within the tumor [564] with more limited side toxicity [565]. It holds the same for PTX, SEVs being more efficient than free PTX and liposomal PTX in inhibiting cancer cell growth [566]. However, developing SEV fusion with liposomes to produce a hybrid exosome (HE) with improved PTX loading capacity and enhanced tumor-targeting ability seems promising for triple-negative TNBC chemotherapy [567]. Loaded SEVs can overcome drug efflux transporter adverse effects, decreasing tumor metastasis compared to controls [568]. Interestingly, SEVs can provide cargo combinational therapy, as shown for the PTX/5-FU association in BC [558]. Very recently, lapatinib-loaded exosomes were developed as a drug delivery system in BC (Değirmenci 2022).

Aside from drug transport, SEVs are natural nucleic acid molecule carriers and can be genetically engineered to deliver specific DNA or RNA molecules. More recently, exosome–liposome hybrid nanoparticles have been developed to deliver the gene editing system CRISPR/Cas9 in mesenchymal stromal cells (MSCs) [569]. SEV vectorization of specific miRNAs has also been used. BC cell proliferation and migration were significantly suppressed when cells were treated with SEVs loaded with miR-142-3p [570] and let7c-5p [571]. EGFR-expressing cells can be targeted with GE11-positive SEVs loaded with miR-let-7a, a tumor suppressor microRNA. The results showed an efficient delivery of SEV cargo and tumor growth inhibition [572]. While miR-134 SEV delivery has been shown to enhance TNBC cells’ drug sensitivity [573], TNBC aggressiveness was suppressed using either miR-381-3p- or miR-145-containing MSC-derived SEVs [574,575]. A synergistic efficacy of co-delivering miR-159 and DOX in SEVs was reported for TNBC therapy [576]. Not only miRNAs can be transferred through SEV delivery. DARS-AS1, a newly reported CUMS-responsive lncRNA, was enriched in TNBC cells and positively correlated with the late clinical stage in patients with TNBC. Treatment with DARS-AS1 siRNA-loaded SEVs substantially slowed CUMS-induced TNBC cell growth and liver metastasis [577].

SEVs can also be used as a new type of tumor vaccine [578]. SEVs have been explored as modulators of the immune response against tumor cells. In BC, treatment with topotecan (TPT, an inhibitor of topoisomerase I) induces BC cells to release SEVs containing DNA that activates dendritic cells (DC) [579]. DCs are antigen-presenting cells that are central to the initiation and regulation of innate and adaptive immunity in the tumor microenvironment. DCs have been shown to secrete antigen-presenting SEVs that coexpress major histocompatibility complex molecules. Such SEVs activate specific cytotoxic T lymphocytes in vivo that can reduce or SEVen suppress tumor growth [580]. SEVs from DCs are likely to initiate an immune response against tumor cells more precisely and accurately than cell therapy and other non-cell-based therapy [581]. Vaccination of transgenic HLA-A2/HER2 mice with a single dose of SEVs from DCs transfected with an adenoviral vector led to activating CD8+ T cell cytolytic functions against BC cells in vitro and reduced tumor growth in vivo [582].

Interestingly, tumor cell-derived SEVs have been shown to have an immunostimulatory effect on antitumor DCs [583]. Such evidence prompts to start engineering DC using targeted-SEV delivery of antigens and adjuvants to DCs, representing a fundamental approach for developing DC vaccines [584]. BC cell line 4T1-derived SEV-mediated transfer of let-7i, miR-155, and miR-142 to DC enhances DC maturation [585].

Cells under different conditions will determine SEV heterogeneity, generating vast and complex combinatorial possibilities. Cell-derived SEVs are generally directed to specific cell types [12]. SEVs derived from hypoxic tumor cells tend to be more easily taken up by hypoxic tumor cells [586]. Thus, to better use SEVs in cancer, engineering SEVs with ligands that can specifically bind to targeted cancer cells is mandatory. Either SEV surface expression of receptor/ligand, antibody/ligand, or microenvironment-specific molecules can be used to modify SEVs. Recently, bioengineered SEVs have been able to specifically bind to HER2 by expressing designed ankyrin repeat proteins (DARPins) on their membrane surface [587]. Directing CD3 and EGFR expressions on SEV membranes was shown to induce cross-linking of T cells and EGFR-expressing BC cells and elicit potent antitumor immunity both in vitro and in vivo [588]. It holds the same when SEVs are engineered through the genetic display of anti-human CD3 and anti-human HER2 antibodies, dually targeting CD3 T cell and BC-associated HER2 receptors. Such SEVs redirect and activate cytotoxic T cells toward attacking HER2-expressing BC cells [589]. Both hyaluronan (HA), the most specific CD44 ligand [590], and CD44 itself, which are both mainly involved in the metastasis process, have been evidenced in BC cells EVs and associated with chemoresistance [506]. High accumulation of HA in the tumor microenvironment leads to an increase in the interstitial pressure and reduced perfusion of drugs. Hyaluronidase, an enzyme that degrades HA, has been engineered into SEV. Hyaluronidase-containing SEVs have been developed and shown to degrade tumor extracellular matrix and enhance the permeability of T cells and drugs within the tumor [591], inhibiting BC metastasis and improving tumor treatment efficiency [592]. Another smart reported strategy was to use HA-engineered SEVs to direct chemotherapy to CD44 expressing BC cells. HA decoration of milk DOX-containing SEVs directs tumor-specific delivery of DOX [593].

Using SEVs as therapeutic vectors in cancer seems very promising, and clinical trials are nowadays being carried out [594]. Unfortunately, breakthroughs still need to occur because of the complexity of handling such new therapeutic methods in vivo. There is also an urgent need to better understand SEV biology and nature to accelerate SEV vectorization in BC patient treatment.

## 8. Conclusions

It is now clearly stated that SEVs exert various biological functions, mainly via delivering signaling molecules that regulate an extensive repertoire of cellular processes. Their role in cancer development seems central as they are significant players in multidirectional signaling between cancer cells and various other ones (from neighboring tumor microenvironment cells at the primary tumor site to more distant ones). It covers every step of BC carcinogenesis up to metastatic dissemination. SEV detection in a large variety of biological fluids represents the future of cancer detection, an easy and reproducible means to identify new specific diagnostic and prognostic biomarkers. SEVs also represent new targets for treatment as their inhibition could limit or stop cancer development. Additionally, these extracellular signaling cargos could be used as specific vectors to convey conventional or innovative therapies to targeted cancer cells.

However, fundamental research is still mandatory to understand SEV function in cancer progression. Although pre-clinical data appear very promising, validation from large clinical trials is needed to support the daily use of SEVs as either tumor biomarkers for monitoring cancer progression and driving treatment decisions or new vectors for specifically targeted treatments.

## Figures and Tables

**Figure 1 ijms-24-07208-f001:**
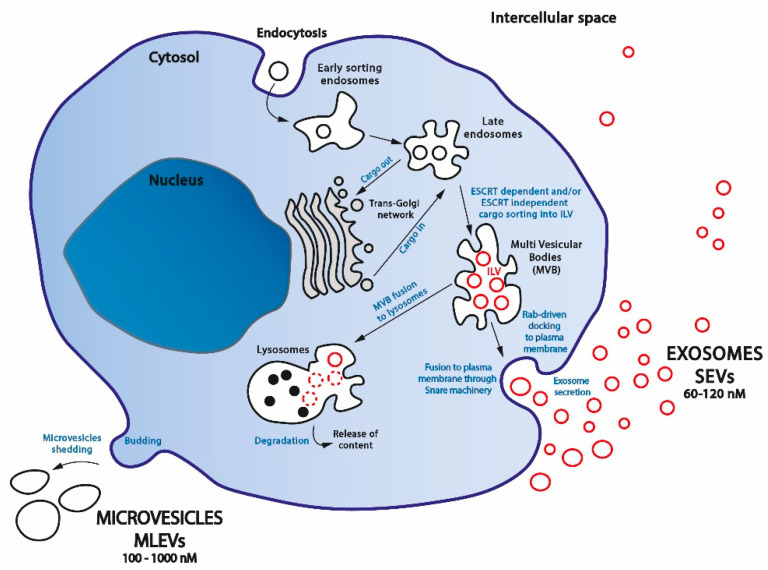
SEVs biogenesis and release. SEVs may have multiple origins. They can originate from plasma membrane budding, which leads to heterogeneous membranous medium-large vesicles (MLEVs) shedding. Small EVs (SEVs, exosomes) originate from the internal budding of plasma membranes giving rise to early endosomes. By complex maturating interactions with the Golgi apparatus (cargo-in/cargo-out), early endosomes become late ones. The membranes of late endosomes form intraluminal vesicles (ILVs), small cargos containing proteins from the plasma membrane, and Golgi as well as nucleic acids. Endosomal cargo sorting was performed through either ESCRT-dependent or -independent routes; the ESCRT complex being the key machinery of protein sorting into SEVs. ILVs are contained in multivesicular bodies (MVBs) that fuse with either plasma membrane (after Rab-driven docking), releasing SEVs in the extracellular space (through Snare complex assembly) or with lysosomes for further internal degradation.

**Figure 2 ijms-24-07208-f002:**
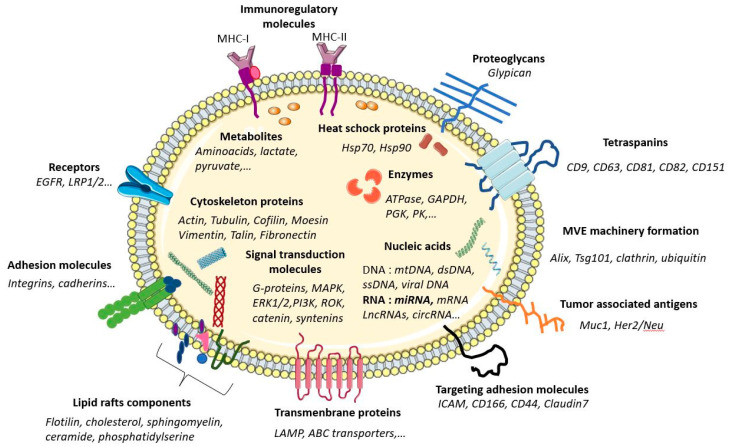
Exosome membrane molecules and their cargo content. Small extracellular vesicles (SEVs) are nano-sized membrane vesicles released by a variety of cell types and are thought to play important roles in intercellular communications. SEVs contain many kinds of proteins, either cytosolic or plasma membrane ones. Transporters, receptors, and signaling proteins, but also enzymes, can be evidenced. Metabolites are also present as well as nucleic acids. Genomic and mitochondrial DNAs and multiple RNAs (mRNAs, miRNA, lncRNA, circRNA, etc.) can be detected. Through the horizontal transfer of these bioactive molecules, SEVS are emerging as local and systemic cell-to-cell mediators of oncogenic information.

**Figure 3 ijms-24-07208-f003:**
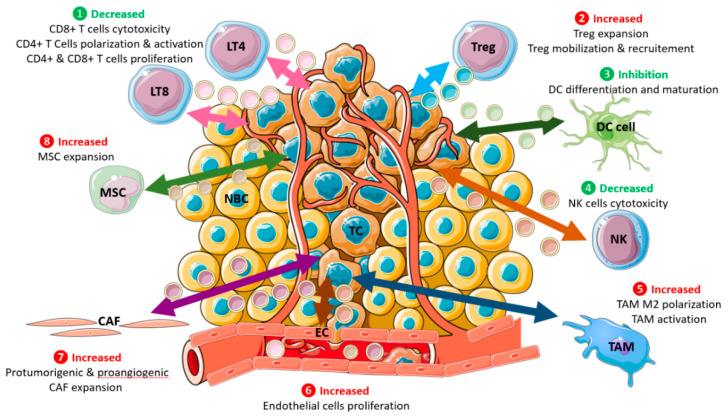
Bidirectional communication between tumor cells and their surrounding environment. The tumor microenvironment (TME) is a complex and dynamic network that includes normal breast (NBC), tumor (TC), cancer-associated fibroblasts (CAFs), mesenchymal stromal (MSC), immune (tumor-associated macrophages TAM), and endothelial cells (EC). TC can bidirectionally signal to each other through SEVs production. TC can produce SEVs that will regulate MSCs’, CAFs’, and TAMs’ differentiation and activity. MSCs as well as TCs can regulate ECs’ activity, especially in hypoxic situations. TAMs, CAFs, and ECs can cooperate to promote angiogenesis. An antitumor immune response is largely modulated by BC cells through either extracellular signaling molecules (cytokines, etc.) secretion or SEVs production and release. BC cells SEVs contain inhibiting or activating molecules that favor target cells expansion, mobilization, and recruitment (CD4+ T cells (LT4), Tregs, and MSCs), polarization and activation (tumor-associated macrophages TAMs M2), and block others (CD8+ T cells (LT8), dendritic cells (DC), and natural killer NK cells).

**Figure 4 ijms-24-07208-f004:**
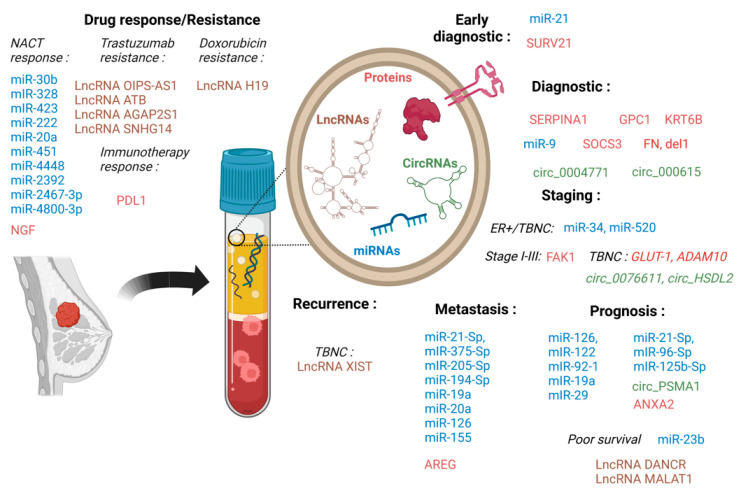
SEVs cargo as relevant breast cancer biomarkers. Among all the molecules present in SEVs, only a subset (proteins, miRNAs, and LncRNAs) have been shown to be of potential clinical value in CRC detection, diagnosis, prognosis, and treatment response evaluation. All referenced markers were found to be differentially expressed in cancer patients and in healthy people: miRs are depicted in blue, LncRNAs in brown, CirRNAs in green, and proteins in pink. NACT: neoadjuvant chemotherapy.This figure was created with BioRender (DT2576K3SR agreement number).

## Data Availability

Not applicable.

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
