# Peer review of "Extracellular Vesicles in Breast Cancer: From Biology and Function to Clinical Diagnosis and Therapeutic Management"

_ijms, 2023, doi:10.3390/ijms24087208_

Round 1

Reviewer 1 Report

The work is scientifically sound, but there were too many abbreviated terms in this manuscript, which I found to significantly affect my comprehension of the whole article. 

Please consider to introduce paragraphs, where necessary, in your manuscript. 

Author Response

We thank reviewer 1 for his/her careful evaluation of our Ms. We agree that our Ms comprised too many abbreviations that can affect reader comprehension. To address this issue, we have withdrawn most of the unnecessary abbreviations. Two redundant abbreviations in all the Ms - BC for breast cancer and SEVs for small extracellular vesicles - have been kept without redefining them throughout the paragraphs. When using them, we systematically defined or redefined all the other abbreviations in the text at the beginning of each paragraph. As requested, for text clarity, we also introduced subsections in our paragraphs when they seemed appropriate. We hope these modifications render our text easier to read for IJMS readers. 

Reviewer 2 Report

The authors thoroughly reviewed the research progress on extracellular vesicles that are related to breast cancer (BC). The review covers a broad range of topics, including the biogenesis and properties of EVs, the EVs’ roles in normal and cancerous breast tissues, EVs as the breast cancer biomarkers for diagnosis, EVs as the targets for BC treatment, and EVs as drug carriers. Overall, this review has a good balance of breadth and depth and provides insights into the remaining challenges and future perspectives of this field. Therefore, this manuscript can be accepted for publication.

Author Response

We sincerely thank reviewer 2 for his/her careful reading of our manuscript and his/her positive evaluation.

Reviewer 3 Report

1. The Title on "in diagnosis and therapeutic management" does not fit the main contents. The management is difficult understanding.

2. Is breast cancer (BC) the first worldwide most frequent cancer in both sexes? The reference is necessary to support it.

3. The information in Figure 1 should be modified more logically containing the details EV genesis and releasing process. Does “EXOXOMES” mean “EXOSOMES”? Do the exosomes come from exosome endocytosis? How is the MLEV budding? Does only MLEV come from budding? How does Golgi apparatus and late endosome cargo in or out? In the figure, the logic relationship of Exosomes, small exosomes and large exosomes should be present clearly. The necessary references on SEVs biogenesis should be added in main text.

4. The authors should renew the updated research achievements on “the essential mechanisms that may account for the 134 combinatorial repertoires of EV cargo and the heterogeneity in cargo compositions across 135 different EV populations and subtypes” after 2019.

5. The title about Figure 2 is suggested as exosome membrane molecules and its cargo content. The information on its membrane molecules should be added in the main text as well.

6. On the EVs production in normal mammary tissue, it is necessary to sum up the EVs oriented from what kinds of the different cell types in normal mammary tissue and what are the specific properties of the EVs product in normal mammary tissue.

7. The part of “Milk is an essential source of EVs” is away from the main topic about EVs genesis from cells and roles between cells. It is suggested to delete.

8. “Bidirectional contribution of breast tumor and microenvironmental cells EVs to BC changes”:  Is this sentence correct in grammar? Check it more carefully.

9. Figure 3 is suggested to be modified to focus on the contribution of EVs on “Bidirectional communication between tumor cells and their surrounding environment”.

10. The part of “EVs deregulation in breast cancer” is suggested to shorten and focus on EVs’ functional mechanisms on breast cancer.

11. Figure 4 is not very clear to highlight the idea of “EVs cargo as relevant breast cancer biomarkers” and the quality is good enough as well.

Author Response

We thank reviewer 3 for his/her careful reading of our manuscript and his/her essential advice. We have answered point by point each of his/her concerns. As it was necessary to highlight with color what has been modified in the text and to add the newly corrected figures according to the reviewer's requirements, our answers have been written in a separate pdf file attached.
We hope these modifications will fulfill reviewer 3 requests and that our Ms in its new form is now suitable for publication in IJMS.  

Round 2

Reviewer 3 Report

1. "BC cells SEVs" (such as at Line 575) should be corrected.

2. Word style in Line 300-328 should be modified.